# GMMIP (v1.0) Contribution to CMIP6: Global Monsoons Model Inter-comparison Project

Tianjun Zhou[1], Andrew Turner[2], James Kinter[3], Bin Wang[4], Yun Qian[5],
Xiaolong Chen[1], Bo Wu[1],Bin Wang[1], Bo Liu[1,6], Liwei Zou[1], BianHe[1]

[1]LASG, Institute of Atmospheric Physics, Chinese Academy of Sciences, Beijing, 100029, China
[2]NCAS-Climate and Department of Meteorology, University of Reading, UK
[3]Center for Ocean-Land-Atmosphere Studies & Dept. of Atmospheric, Oceanic & Earth Sciences, George Mason University, USA
[4]University of Hawaii, USA
[5]Atmospheric Sciences & Global Change Division, Pacific Northwest National Laboratory, Richland, USA
[6]Graduate University of the Chinese Academy of Sciences, Beijing, 100049, China

*Correspondence to*: Tianjun Zhou (zhoutj@lasg.iap.ac.cn)

**Abstract.** The Global Monsoons Model Inter-comparison Project (GMMIP) has been endorsed by the panel of Coupled
Model Inter-comparison Project (CMIP) as one of the participating MIPs in the sixth phase of CMIP (CMIP6). The focus of GMMIP is on monsoon climatology, variability, prediction and projection, which is relevant to four of the "Grand Challenges" proposed by the World Climate Research Programme. At present, 21 international modelling groups are committed to joining GMMIP. This overview paper introduces the motivation behind GMMIP and the scientific questions it intends to answer. Three tiers of experiments, of decreasing priority, are designed to examine: (a) model skill in simulating
the climatology and interannual-to-multidecadal variability of global monsoons in SST-forced experiments of the historical climate period; (b) the roles of the Interdecadal Pacific Oscillation and Atlantic Multidecadal Oscillation in driving variations of the global and regional monsoons; and (c) the effects of large orographic terrain on the establishment of the monsoons. The outputs of the CMIP6 DECK, "historical" simulation and endorsed MIPs will also be used in the diagnostic analysis of GMMIP to give a comprehensive understanding of the roles played by different external forcings, potential
improvements in the simulation of monsoon rainfall at high resolution and reproducibility at decadal time scales. The implementation of GMMIP will improve our understanding of the fundamental physics of changes in the global and regional monsoons over the past 140 years and ultimately benefit monsoons prediction and projection in the current century.

**Key Words** Monsoon, anthropogenic forcing, internal variability, interdecadal variability, Interdecadal Pacific Oscillation, Atlantic Multidecadal Oscillation, interannual variability, elevated heating

**1 Introduction**

Changes in the precipitation and atmospheric circulation of the regional monsoons are of great scientific and societal importance owing to their impacts on more than two-thirds of the world's population. Prediction of changes to monsoon

rainfall in the coming decades is of deep societal concern and vital for infrastructure planning, water resource management, and sustainable agricultural and economic development, often in less developed regions.

The dominant monsoon systems defined by precipitation characteristics include the Asian, Australian, Northern and Southern African, the North American, and the South American monsoons (Wang, 1994; Wang and Ding, 2008; Fig. 1). Each system generally has its own unique characteristics in terms of the evolution, variability and impacts due to its indigenous land-sea configuration and the particular atmosphere-ocean-land interaction involved. At the same time, the regional monsoons have in common the fundamental driving factors of temperature and pressure gradients, and they are bounded by the global divergent circulation necessitated by mass conservation as they evolve through the season (Trenberth et al., 2000). The global monsoon represents the dominant mode of the annual variation of precipitation and circulation in the global tropics and subtropics (Wang and Ding, 2008) and as such, the global monsoon is a defining feature of Earth's climate. On the annual time scale, the global monsoon is a planetary scale circulation system with a seasonal reversal of the three-dimensional monsoon circulation that is accompanied by the migration of the monsoon rainfall zones. However, it remains debatable to what extent and at which time scales the global monsoon – defined as the regional monsoons acting together – can be viewed as a major mode of climate variability (Wang et al., 2014). To facilitate the discussion, we use "global monsoon" to regard all the monsoon domains as a whole and a single phenomenon to highlight the integrated role of monsoons in global hydrological cycle, whereas we use "global monsoons" to highlight the regional features of different monsoon domains ver the globe.

To what extent can internal feedback processes in the climate system drive the interannual variations of global monsoon precipitation? Wang et al. (2012) have shown that from one monsoon year (defined as May to the next April) to the next, most continental monsoon regions, separated by vast areas of arid trade winds and deserts, vary in a cohesive manner driven by the El Niño-Southern Oscillation (ENSO). On decadal time scales, numerous studies have investigated the linkage between regional monsoons and other major modes of climate variability. For instance, the Australian summer monsoon was linked to the Interdecadal Pacific Oscillation (IPO; Power et al., 1999a); the Indian summer monsoon precipitation has a correlation with the North Atlantic Oscillation (NAO) (Goswami et al., 2006) and the IPO (Meehl and Hu, 2006); the East Asian summer monsoon is related to the Atlantic Multidecadal Oscillation (AMO; Enfield et al., 2001; Lu et al., 2006) and the Pacific Decadal Oscillation (PDO; Mantua and Hare, 2002; Li et al., 2010; Qian and Zhou, 2014; Zhou et al., 2013); the variability of the west African and North American monsoons is related to the AMO (Sutton and Hodson, 2005; Zhang and Delworth, 2006; Gaetani and Mohino, 2013); and the African monsoon system is sensitive to inter-hemispheric SST variability in the Atlantic (Folland et al., 1986; Hoerling et al., 2006). Many decadal and interdecadal variations of regional monsoons have been identified, with differing periodicity and phase change points (Yim et al., 2014; Chen and Zhou, 2014; Lin et al. 2014). While these concepts can be collated and simplified by considering processes controlling the position of the zonal mean ITCZ (Schneider et al., 2014), a coherent global structure and the underlying causes of global monsoon interdecadal variability have yet to be widely studied.

The combination of changes in monsoon area and rainfall intensity has led to an overall weakening trend of global land monsoon rainfall since the 1950s (Wang and Ding, 2006; Zhou et al., 2008a). This decreasing tendency is dominated by the African and South Asian monsoons, as shown by the significant decreasing tendencies of both rainfall intensity and monsoon coverage area (Zhou et al., 2008b). Beginning in the 1980s, however, the Northern Hemisphere global monsoon precipitation has had an upward trend (Wang et al., 2012). These studies of the trends in relatively short precipitation records have not been able to confirm whether these trends are part of longer-period fluctuations. Recently, Wang et al. (2013) studied coherent interdecadal variations of the Northern Hemisphere summer monsoon (NHSM) by using the NHSM circulation index (defined by the vertical shear of zonal winds between 850 hPa and 200 hPa averaged in $0°-20°N, 120°W-120°E$). The NHSM circulation index is highly correlated with the NHSM rainfall intensity over the modern record (r=0.85 for 1979-2011). They demonstrated that the NHSM circulation has experienced large-amplitude multidecadal fluctuations since 1871, primarily attributed to a mega-ENSO (a leading mode of interannual-to-interdecadal variation of global sea surface temperature) and the AMO. Only about one third of the recent increasing trend in the NHSM rainfall since 1979, when measured across the whole northern hemisphere, was attributed to anthropogenic warming.

How forcing agents, including both of the anthropogenic and natural, impact global monsoons changes is another important but tough question. Dynamical and thermodynamical changes of monsoon rainfall could cancel each other to some extent under greenhouse gases (GHG) forcing (Cherchi et al., 2011; Endo and Kitoh, 2014; Chen and Zhou, 2015).However, the relative contributions of these two processes to observed global monsoon rainfall changes due to anthropogenic GHG forcing are still unknown. The interaction of aerosol forcing with monsoon dynamics may alter the redistribution of energy in the atmosphere and at the Earth's surface, thereby changing the monsoon-related water cycle and climate (Lau et al., 2008). Aerosols may reduce surface solar insolation, thus weakening the land-ocean thermal contrast and modifying the formation and development of monsoons. Many mechanisms have been proposed in the past two decades regarding the impact of aerosols on monsoon circulation and precipitation. These mechanisms are complicated by the feedbacks with large-scale moist environmental dynamics, so large uncertainties still remain (Qian and Giorgi, 1999; Menon et al., 2002; Qian et al., 2002, 2009, 2011). The aerosol-monsoon interaction has attracted rapidly increasing interest in the global climate modelling community. The relative importance of aerosol forcing and global warming to observed trend of monsoon rainfall, for example the decreasing of Indian rainfall in the recent decades, also need to be clarified (Bollassina et al., 2011; Annamalai et al., 2013; He et al., 2016).Understanding the mechanisms of precipitation changes in the global monsoon system and identifying the roles of natural and anthropogenic forcing agents have been central topics of the monsoon research community (Cook et al., 2013; Liu et al., 2013; Song et al., 2014; Polson et al., 2014; Guo et al., 2015).

While all monsoons are associated with large-scale cross-equatorial overturning circulations, major differences in the characteristics of the regional monsoons arise because of the different orography and underlying surface as well as the external forcing. This is most apparent for the Asian region, due to the Tibetan – Iranian Plateau, Himalayan mountains and strong anthropogenic forcing from aerosol emissions and land-use change. The highlands may act as a physical barrier that isolates the heat and moisture south of the Himalaya and a high-level heat source (pump) that directly drives the monsoon

circulation through meridional thermal contrast (Yeh, 1957; Flohn, 1957; Yeh and Wu, 1998; Yanai and Wu, 2006). However, the relative role of the two effects deserve more discussion (Boos and Kuang, 2010, 2013; Wu et al., 2012; Qiu et al., 2013).

Climate models are useful tools in climate variability and climate change studies. However, the performance of current state-of-the-art climate models is very poor and needs to be greatly improved over the monsoon domains (Cook et al., 2012; Kitoh et al., 2013; Wang et al., 2005; Zhou et al., 2009a; Sperber et al., 2013; Song and Zhou, 2014ab). As one of the endorsed MIPs in the sixth phase of the Coupled Model Inter-comparison Project (CMIP6) (Eyring et al., 2016), the Global Monsoons Model Inter-comparison Project (hereafter GMMIP) aims to improve our understanding of physical processes in global monsoon systems by performing multi-model inter-comparisons, ultimately to work towards better simulations of the mean state, interannual variability and long-term changes of the global monsoons. The contributions of internal variability (IPO and AMO) and external anthropogenic forcing to the historical evolution of global monsoons in the 20th and 21st century will also be addressed.

GMMIP aims to answer four primary scientific questions:

(1) What are the relative contributions of internal processes and external forcing that are driving the historical evolution of monsoons over the late 19th through early 21st centuries?

(2) To what extent and how does atmosphere-ocean interaction contribute to the interannual variability and reproducibility of monsoons?

(3) How can high resolution and associated improved model dynamics and physics help to reliably simulate monsoon precipitation and its variability and change?

(4) What is the effect of the orography of the Himalaya/Tibetan Plateau on the development and maintenance of the Asian monsoon? Similarly, what is the impact of orography elsewhere on other regional monsoons?

By focusing on addressing these four questions we expect to deepen our understanding of the capability of models to reproduce the monsoon mean state and its natural variability as well as the forced response to natural and anthropogenic forcing, which ultimately will help to reduce model uncertainty and improve the credibility of models in projecting future changes in the monsoon. The coordinated experiments will also help advance our physical understanding and prediction of monsoon changes.

Due to the uncertainties in physical parameterizations in current models, particularly in convection schemes (Chen et al., 2010), the best way to address the above questions is through a multi-model framework in order capture the range of possible responses to forcing. The multi-model database to be produced for CMIP6 (Eyring et al., 2016), in conjunction with the GMMIP experiments will provide an opportunity for advancement of monsoon modelling and understanding. GMMIP will also contribute to the Grand Challenges of the World Climate Research Programme (WCRP) and address them in the following way:

(1) Water Availability

The water resources in global monsoon domains are greatly affected by the anomalous activities of monsoons. The summer monsoons produce more than 80% of the annual rainfall in some areas, e.g., in India, Africa and Australia, and the percentage is more than 60% averaged across all global monsoon regions (Fig. 2). Understanding the mechanisms of monsoon variability on interannual and longer time scales as posed by GMMIP will lead to improvement of monsoon

prediction and projection and provide useful information for policymakers in water availability-related decision making.

(2) Climate Extremes

Extreme events such as mega-droughts and flooding are frequent occurrences in monsoon domains. GMMIP will allow the impact of changing lower boundary forcing on the statistics of extreme events to be examined in a consistent manner.

(3) Clouds, Circulation and Climate Sensitivity

A reasonable simulation of monsoon circulation is a prerequisite for a successful simulation of monsoon precipitation (e.g., Sperber et al., 2013). At the same time, tropical precipitation is strongly dependent on convection, with monsoon precipitation biases very sensitive to convective parameterizations and therefore clouds. These parameterizations also lead to large uncertainties in climate sensitivity (e.g., Stainforth et al. 2005). By comparing the performance of climate models with relatively high and low resolutions, and model simulations with and without air-sea interaction processes, GMMIP will

attempt to link monsoon precipitation simulation with the fidelity of the large-scale circulation and latest remote sensing estimates of clouds.

## 2 Participating models

So far 21 international modelling groups have committed to contributing to GMMIP (as shown in Table 1). The diversity of the groups from different countries and regions demonstrates that the global monsoons topic appeals to a wide range of

modelling and research communities. The models with various structures, physical parameterizations, resolutions etc. will provide a large sample size to help reveal the causes of monsoon variability on interannual and longer time scales in the climate system. Based on the experimental protocol (see Section 3), both atmosphere-only and fully coupled ocean-atmosphere versions of these models will be used.

## 3 Experimental protocol

Based on the priority level of proposed scientific questions, the main experiments of GMMIP, which are summarized in Table 2, are divided into ***Tier-1, Tier-2, and Tier-3 of decreasing priority*** (Fig. 3). In order to diagnose internal variability, at least 3 members integrated from different initial conditions are required for Tier-1 and Tier-2 experiments. Pending the availability of computer resources at GMMIP-committed climate-modelling centres, realizations with more than 3 members are encouraged.

### 3.1 Tier-1: Extended AMIP experiment

The Tier-1 experiments are extended AMIP runs from 1870 to 2014. This is the entry card for GMMIP. All external forcings (solar, aerosol, GHGs, etc.) should be derived from those used in the Historical simulation of the CMIP6 fully coupled model. This will allow a direct comparison of the Historical simulation and extended AMIP run, to determine the importance of SST variability to long and short-term trends in the monsoon circulations and the associated precipitation. The boundary conditions for sea-surface temperature and sea ice are derived from a merged version of the Hadley Centre sea-Ice and SST (HadISST) and Optimum Interpolation Sea Surface Temperature (OISST) data sets (Hurrell et al., 2008), which can be downloaded from the PCMDI website[1].

### 3.2 Tier-2: Decadal mode relaxation experiments

The Tier-2 experiments are initialized from "historical" run year 1870 and integrated up to year 2014 with historical forcings. Additionally, the variation in the Tropical Pacific and North Atlantic SST are restored to the observation in the "hist-resIPO" and "hist-resAMO" runs, respectively. The Tier-2 "hist-resIPO" (historical anthropogenic forcing plus restoring IPO SST) run is a pacemaker historical coupled climate simulation that includes all forcings as in the CMIP6 historical experiment, but with SST restored to the model climatology plus observed historical anomaly in the tropical lobe of the Interdecadal Pacific Oscillation (IPO; Power et al., 1999; Folland et al., 2002) domain (20°S-20°N, 175°E-75°W). This relaxation is applied with weight=1 in the inner box (15°S-15°N, 180°-80°W) and linearly reduced to zero in the buffer zone (zonal and meridional ranges are both 5°) from the inner to outer box (Fig. 4a). There are several restoring methods to realize such "pacemaker" simulations (see the Appendix I). To ensure stability during integration, we recommend nudging to the specified SST described above with a 10-day time scale (see the Appendix I for technical details).

Similarly, the Tier-2 "hist-resAMO" (historical anthropogenic forcing plus restoring AMO SST) run is a pacemaker historical coupled climate simulation that includes all forcings but with the SST restored to the model climatology plus observational historical anomaly in the Atlantic Multidecadal Oscillation (AMO; Enfield et al., 2001; Trenberth and Shea, 2006) domain (0°-70°N, 70°W-0°). The restoration is fully applied in the inner box (5°N-65°N, 65°W-5°W), and linearly reduced to zero in the buffer zone (zonal and meridional ranges are both 5°) from the inner to outer box (Fig. 4b).

### 3.3 Tier-3: Orographic perturbation experiments

The Tier-3 experiments is generally the same as the "amip" run in the CMIP6 DECK covering 1979-2014 except that some key topographies or air-land sensible heat flux are modified. The aim of the "orographic perturbation" is to understand quantitatively the regional response to the orographic perturbation from both the thermal and dynamical aspects. The results will be very helpful to understand the topography effect on the atmosphere and associated physical processes locally and quantitatively, such as the distribution, intensity, and frequency changes in the precipitation over wide monsoon regions. In

---

[1]http://www-pcmdi.llnl.gov/projects/amip/AMIP2EXPDSN/BCS/amipbc_dwnld.php

the Tier-3 "amip-TIP" run (viz. no Tibetan - Iranian Plateau) run, following Boos and Kuang (2011, 2013) and Wu et al. (2007, 2012), the topography of the Tibetan-Iranian Plateau (hereafter TIP, see Table 2 for detailed descriptions) in the model is modified by leveling off the TIP to 500m, with other surface properties unchanged (*Asia* region in Fig. 5 and details seen in Appendix II). Other settings of the integration are the same as the standard DECK AMIP run. This experiment represents perturbations to both thermal and mechanical forcing of the TIP with respect to the standard DECK AMIP run. In an ensemble of experiments comprising the Tier-3 "amip-hld" run (viz. no HighLanDs) group, the topography of the East African Highlands in Africa (after Slingo et al., 2005), the Sierra Madre in North America and the Andes in South America is modified by setting surface elevations to 500m in those respective regions (Fig. 5).

The sensible heat over the elevated topography is regarded as the main driver of the behaviour of the low level atmosphere and possibly also the upper troposphere and lower stratosphere (Wu et al., 2016). To examine the importance of elevated heating in monsoon from perspective of multi-model comparison, in the Tier-3 "amip-TIP-nosh"run (viz. Tibetan - Iranian Plateau - no sensible heating), the surface sensible heat flux at elevations above 500m over the TIP is not allowed to heat the atmosphere, i.e., the vertical temperature diffusion term in the atmospheric thermodynamic equation at the bottom boundary layer is set to zero (Wu et al., 2012; details seen in Appendix II). The atmospheric component will not see the surface upward sensible heat flux (zero), whereas the land component is as usual. Other settings of the integration are the same as the standard DECK AMIP run. The differences between the standard DECK AMIP run and the amip-TIP-nosh are considered to represent the removal of TIP thermal forcing only and thus the circulation pattern of amip-TIP-nosh reflects the impacts of mechanical forcing.

### 3.4 Experiment outputs

The recommended output variables are listed in Appendix III.

### 4 Connection with DECK, Historical Simulation and endorsed MIPs

The Tier-1 experiment of GMMIP, i.e. the extended AMIP, uses the same resolution as in the DECK (Eyring et al., 2016). The amip-hist specifies external forcings that are consistent with those from the same model's CMIP6 Historical Simulation over the 1870-2014 period. To comprehensively investigate the proposed GMMIP scientific questions, such as the impact of high resolution and roles of different forcing agents, the output from other related MIPs will be used in the diagnostic analysis of GMMIP as described below.

### 4.1 DECK and Historical Simulation

The pre-industrial control simulations from each modelling group's DECK experiments will be used to study the relation between global monsoon and IPO/AMO at decadal time scale. Comparing the control simulation (constant forcing) with the GMMIP Tier-2 decadal mode relaxation experiments in which all historical forcings are added will then allow us to find

which parts of apparent decadal variations in the monsoons are caused by underlying SST, and which are more forced by externally driven sources, such as volcanic emissions. The CMIP6 historical simulations will also be used to examine the response of the global monsoon to external forcings such as anthropogenic GHG and aerosol emissions. The results of CMIP6 historical simulation will be compared with those of hist-resIPO and hist-resAMO in Tier-2 to identify the relative

contributions of external forcing and apparently internal modes of variability (IPO/AMO).

## 4.2 DAMIP (Detection and Attribution MIP)

Several DAMIP experiments are useful to GMMIP. The histALL (enlarged ensemble size of historical all-forcing runs in DECK), histNAT (historical natural forcings-only run), histGHG (historical well-mixed GHG-only run), and histAER experiments (historical anthropogenic-Aerosols-only run) of DAMIP will be used in the analysis of changes in global

monsoons dating back to 1870.

Analyzing combinations of the histALL, histNAT and histGHG ensembles will allow us to understand the observed evolution of global monsoon precipitation and circulation changes since 1870 in the context of contributions from GHG, the other anthropogenic factors and natural forcing. The contributions of these external forcings to global monsoon changes will be compared to those from modes of internal variability such as the IPO and AMO.

## 4.3 HighResMIP (High Resolution MIP)

The Tier-1 experiments of HighResMIP, which consist of AMIP runs with a minimum horizontal resolution of 25-50 km, will be used to compare with standard resolution control configurations and examine the added benefit, if any, of high-resolution models in reproducing both the mean state and year-to-year variability of global monsoons. It should be noted that the boundary conditions (both of SST and sea ice) used to build the AMIP experiments of HighResMIP is a new dataset with

daily time frequency (Haarsma et al., 2016), which may make differences when comparing with standard AMIP forced by monthly datasets.

The Tier-2 experiments of HighResMIP, which are coupled runs consisting of pairs of both historic runs and control runs using fixed 1950s forcing including anthropogenic GHG concentrations and aerosol forcing, will be used in the analysis of climatology and variability of global monsoons, which aims to understand the role of air-sea interaction in modulating the

simulation skill of the monsoon mean state and year-to-year variability. The anthropogenic aerosols are required to be prescribed in HighResMIP experiments following a standard method in CMIP6 DECK (Haarsma et al., 2016), rather than interactive aerosol processes embedded in atmosphere general circulation models (AGCMs). Different ways to deal with aerosols could lead to different aerosol distributions as well as aerosol forcings, which should be taken in consideration when comparing with GMMIP experiments.

**4.4 VolMIP (Volcanic forcing MIP)**

The Tier-1 experiment of the short set of VolMIP simulations is designed to create a large ensemble of short-term simulations of the 1991 Pinatubo eruption, using the same volcanic forcing recommended for the CMIP6 Historical simulation. It will be used in comparison with observations to understand the global monsoon response to injection of stratospheric aerosols over the tropics and to study impact mechanisms on global monsoon precipitation and circulation changes. Via its ensemble design, VolMIP can address the substantial uncertainty associated with the effects of volcanism during the historical period.

**4.5 DCPP (Decadal Climate Prediction Project)**

The outputs of DCPP near-term climate prediction experiments will be used to assess the skill of global monsoons in initialized decadal climate prediction. The C-component of DCPP is similar to the Tier-2 experiment of GMMIP but focuses on a shorter time period starting from 1950 (Boer et al. 2016). The outputs will be used to add to the ensemble size of pacemaker experiments from GMMIP Tier-2 during the 1950-2014 period.

**4.6 CORDEX (international Coordinated Regional Downscaling Experiment)**

In the core framework of CORDEX phase 2 (CORDEX2 hereafter), a core set of regional climate models (RCMs) downscales a core set of GCMs over all or most CORDEX domains at 10-20 km resolutions (Gutowski Jr., et al. 2016). The comparisons of CORDEX2 historical climate downscaling with the driving GCMs historical simulations, will give insight into the importance of model resolution and the added value of RCMs in the simulation of climatology and variability of global monsoon, especially the global land monsoon. A comparison of CORDEX2 evaluation framework experiments forced with daily mean SST to HighResMIP Tier 1 runs over global monsoon domains will provide information on the similarities and differences of the added values derived respectively from high resolution global models and regional climate models.

**5 Analysis plan**

The analysis plan will focus on the scientific objectives of GMMIP. We list the key scientific questions that we hope that the community will be able to answer following the implementation of GMMIP below.

**5.1 Understanding the changes of global monsoons since the 1870s**

We will examine whether decadal and multi-decadal variability of local monsoon systems and coherent changes of the global monsoon can be reproduced in the amip-hist experiment. Firstly, the skill of reproducing interannual and interdecadal changes in the regional monsoons will be compared with long-term observed records in local monsoon regions, such as using the All-India Rainfall index from 1870 (Parthasarathy et al., 1994) and the CRU global land precipitation from 1901

(Harris et al., 2014; Zhang and Zhou, 2011). The simulated monsoon circulation can be compared with 20CR and ERA20C reanalysis, which are also derived from AGCM simulations driven by observational SST, with surface pressure (marine wind additionally used in ERA20C) records are assimilated (Compo et al., 2011; Poli et al., 2016).

Secondly, the interannual variability of the monsoon systems has experienced dramatic interdecadal variations during past 60 years (e.g., since the 1950s to present, Wang and Ding 2006). The amip-hist results will be used to explore whether similar modulations occurred during the past 150 years, and what mechanisms are responsible for them.

Thirdly, the contributions of apparently internal variability modes (IPO and AMO) to global monsoon variability and the role of air-sea interaction will be evaluated based on the hist-resIPO and hist-resAMO experiments of Tier-2. Combined with CMIP6 DECK and DAMIP experiments, the roles of external forcing (GHG, aerosol, solar, etc.) and internal variability can be quantified. The impact of tropical volcanic eruption on the global monsoons can be explored specifically by analyzing VolMIP. Current state-of-the-art climate models still show bias in the simulation of monsoon (Sperber et al., 2013). We acknowledge that attention should be paid to the model bias in the analysis of model outputs, although multi-model ensemble/intercomparison approach is a useful way to reduce the uncertainty related to model bias.

## 5.2 Effect of air-sea interaction on interannual variability of precipitation in the global monsoons

Previous studies have noted that AGCM simulations with specified SST generally have low skill in simulating the interannual variation of the summer precipitation over global monsoon domains, especially the East Asian-western North Pacific summer monsoon domain (Wang et al., 2005). It is noted that in the real world the precipitation is negatively correlated with underlying SST in the western North Pacific monsoon domain, which is not reproduced by the AMIP runs (Wang et al., 2005). The deficiency of the AMIP simulations can be partially attributed to the exclusion of air-sea interactions (Song and Zhou, 2014b). Comparison between the Tier-1 and Tier-2 experiments of GMMIP can provide information about how the air-sea interactions influence the monsoon simulations on the interannual and interdecadal time scales in different monsoon domains. However, mean state tropical SST biases prevalent in coupled models are also known to affect the accurate connection of monsoon interannual variability with teleconnected drivers such as ENSO (Turner et al., 2005).

## 5.3 Measuring improvement in the global monsoons with high resolution modelling

Monsoon rainbands such as the Mei-yu/Baiu/Changma front usually have a maximum width of about 200 km (Zhou et al., 2009b). Climate models with low or moderate resolution are generally unable to realistically reproduce meso-scale cloud clusters embedded in the rainbands, thus partly leading to biases in the mean state, variability of monsoon precipitation and the northward propagation of these rainbands. We will examine the performance of high-resolution models in reproducing both the mean state and year-to-year variability of global monsoons. High-resolution rain-gauge observations and satellite precipitation products will be used to evaluate model performance.

**5.4 Effects of large orographic terrain on the regional/global monsoons**

The influence of the large-scale orography on the Asian summer monsoon includes both mechanical and thermal forcing. Various mechanisms have been suggested concerning the topographic effects; however, an overarching paradigm delineating the dominant factors determining these effects and the strength of impacts needs further study. We will analyze the Tier-3 experiments to provide a benchmark of current model behaviour in simulating the impact on the monsoon of the Tibetan-Iranian Plateau (TIP, as well as surrounding regions of significant orography, see Table 2 for detailed descriptions) so as to stimulate further research on the thermodynamic and dynamic influence of the TIP on the monsoon. In particular the relative contributions of thermal and orographic mechanical forcing by the TIP on the Asian monsoon will be addressed. We will extend the studies from the TIP to other highlands including highlands in Africa, North America and South America.

**5.5 Aerosol-monsoon interaction**

While aerosol-cloud interaction (ACI) effects are partially incorporated in GCMs with various levels of complexity, the aerosol-radiation interaction (ARI) effect, which is believed to have more explicit impact on land-sea thermal contrast by reducing the surface solar insolation, is fully incorporated in most of CMIP6 models. To investigate the aerosol impacts on monsoon climate including both local forcing and remote forcing effects, we will examine the responses of climate models to natural (solar variability and volcanic aerosols) and anthropogenic (GHGs and aerosols) forcings based on DECK and DAMIP experiments. In particular, we will quantify and compare the separate climatic response of natural vs. anthropogenic forcing, and aerosol vs. GHG forcing, over the global monsoon area (e.g., Song et al., 2014). We will analyze how different forcings influence the general circulation and precipitation characteristics, such as extreme events, shift of precipitation spectrum, and diurnal cycle etc.

**6 Concluding remarks**

Several regions of the world are dominated by a monsoon-like cycle of rainy and dry seasons, which have a profound influence on ecosystems and human agriculture, economy and culture. Diabatic heating released during monsoon rainfall and its effect on the tropical and global atmospheric circulation extend the influence of monsoons globally. It is critical, then, to improve our understanding of the global monsoon, both in terms of better predicting the monsoon on short time scales and developing better projections of how the monsoon is likely to change in the future. The set of numerical experiments proposed for the GMMIP project, in conjunction with the experiments of partner MIPs such as DAMIP, HighResMIP, VolMIP, DCPP, and CORDEX, will help answer some fundamental scientific questions about the global monsoon and will help provide guidance about the future of monsoons as the planet's climate changes. It is also hoped that the GMMIP will provide a good platform for the international climate modelling community in the collaboration of monsoon studies.

**Data Availability**

The model output from the GMMIP simulations described in this paper will be distributed through the Earth System Grid Federation (ESGF) with digital object identifiers (DOIs) assigned. As in CMIP5, the model output will be freely accessible through data portals after registration. In order to document CMIP6's scientific impact and enable ongoing support of CMIP, users are obligated to acknowledge CMIP6, the participating modelling groups, and the ESGF centres (see details on the CMIP Panel website at http://www.wcrp-climate.org/index.php/wgcm-cmip/about-cmip). Further information about the infrastructure supporting CMIP6, the metadata describing the model output, and the terms governing its use are provided by the WGCM Infrastructure Panel (WIP) in their invited contribution to this Special Issue. Along with the data itself, the provenance of the data will be recorded, and DOI's will be assigned to collections of output so that they can be appropriately cited. This information will be made readily available so that published research results can be verified and credit can be given to the modelling groups providing the data. The WIP is coordinating and encouraging the development of the infrastructure needed to archive and deliver this information. In order to run the experiments, datasets for natural and anthropogenic forcings are required. These forcing datasets are described in separate invited contributions to this Special Issue. The forcing datasets will be made available through the ESGF with version control and DOIs assigned. In addition, observational SST and sea ice data are also required. These data are derived from a merged version of the HadISST and OISST datasets, which can be downloaded from the PCMDI website.

**Appendix I: Restoring methods used in the "pacemaker" experiment**

Owing to the difference in model formulation and the difficulty that some users may face in operating pacemaker experiments in coupled models, we offer a choice of three recommended methods for restoring the SST in the hist-resIPO experiments. The first method is recommended for hist-resAMO experiments.

(a) Restoring model SST in every model time step to the corresponding constructed daily SST with a time scale $\tau$. To reduce model drift, the constructed SST is the sum of the model daily climatological SST with seasonal cycle for the period of 1950-2014 from the corresponding historical coupled simulation and the daily SST anomalies in the observation, which are interpolated from the raw observed monthly SST anomalies with the seasonal cycle for the same period removed. We suggest to use the AMIP SST to calculate the observational anomalies, consistent with Tier-1 experiment.

$$\frac{dT}{dt} = \text{Original trend terms} + \frac{(\overline{T_*}+T')-T}{\tau}. \tag{1}$$

Here $T$ denotes the SST and the asterisks represent model-diagnosed values. The prime (bar) refers to the anomaly (climatology). Here the anomaly is based on AMIP SST, while the model's climatology refers to the seasonally evolved daily mean during 1950-2014 based on historical simulations. For the hist-resIPO (hist-resAMO) experiments, the restoring timescale is $\tau = 10$ days ($\tau = 60$ days). The reason for a short timescale (10 days) used in hist-resIPO is that we also aim to study the decadal difference of interannal variability. Too weak restoring may reduce the observed interannual signal.

(b) Prescribing the SST directly in the first layer of ocean component. In the restoring regions, the SST is equal to the model climatology plus the observational anomaly using formula (2).

$$T = (1 - \alpha)T_* + \alpha\left(\overline{T_*} + T'\right). \tag{2}$$

In the inner box (Figure 4), the weighting term $\alpha = 1$, then $\alpha$ is linearly reduced to zero in the buffer zone between inner and outer boxes.

(c) Prescribing the surface net heat flux to restore the SST indirectly. This method has been used in Kosaka and Xie (2013) for hist-resIPO like experiment. In the restoring regions, the heat flux is restored using formula (3). Here $\alpha$ has the same meaning as that described in (2).

$$F = F_* + \alpha\left(\frac{c_p\rho D}{\tau}\right)(T' - T_*'). \tag{3}$$

Here $F$ denotes the heat flux; $c_p$ denotes constant-pressure specific heat of sea water; $\rho$ is the density of the sea water. For the hist-resIPO (hist-resAMO) experiments, the typical depth of the ocean mixed layer is $D = 10$ m ($D = 50$ m) and the restoring timescale is $\tau = 10$ days ($\tau = 60$ days). $T'$ and $T_*'$ are the SST anomalies of AMIP and model SST, respectively, relative to the climatology during 1950–2014. The model's climatology is calculated from the historical simulation. The anomalies instead of full SST used here is to reduce possible drift. A similar restoring method is recommended in DCPP Component C experiments (C1.9 and C1.10) except that full SST is used (Boer et al., 2016).

**Appendix II: Orography and sensible heating modification methods used in the Tier-3 experiment**

(a) "Orographic perturbation": The orography height $H$ is modified in the model with the criterion of $H = 500$ m when $H > 500$ m. The modified Asian region is a polygon region. Coordinates of the polygon corners are as follows: longitude (from west to east), 25 °E, 40 °E, 50 °E, 70 °E, 90 °E and 180 °E; latitude (from south to north), 5 °N, 15 °N, 20 °N, 25 °N, 35 °N, 45 °N and 75 °N. The East African Highlands is in a polygon region. Coordinates of the polygon is as follows: longitude (from west to east), 27 °E and 52 °E; latitude (from south to north), 17 °S, 20 °N and 25 °N, 35 °N. Sierra Madre domain is 120 °–90 °W, 15 °–30 °N. Andes domain is 90 °–60 °W, 40 °S-10 °N. The regions are depicted as the black contour in Fig.5. The domain details of orography to be modified are also seen in Table 2.

(b) "Remove sensible heating": The vertical diffusion heating at the atmospheric model bottom in the Planet Boundary Layer scheme is set to zero ($\partial T / \partial z = 0$) in each step of the model's integration. Here the $T$ denotes the temperature tendency due to heating, and $\partial / \partial z$ denotes the vertical diffusion at the lowest level of the atmospheric model.

**Appendix III: Description of the recommended output**

The following tables list the recommended variables at three time frequencies. There are three priority levels. Smaller number means higher level. Variable names refer to those in the CMIP5. The monthly data is used to analyze the long-term

trend and variability from interannual to multi-decadal time scales. The daily and 6-hourly data is used to study intraseasonal phenomenon and extreme climate.

Table A1 Recommended GMMIP output. The variables in ocean and sea ice realms are only for Tier-2 experiments.

| Output name | Description | Priority | | |
|---|---|---|---|---|
| | | Monthly | Daily | 6-hourly |
| TOA fluxes | | | | |
| rlut | TOA outgoing longwave radiation | 1 | 2 | 3 |
| rsdt | TOA incident shortwave radiation | 1 | 3 | |
| rsut | TOA outgoing shortwave radiation | 1 | 3 | |
| rlutcs | TOA outgoing clear-sky longwave radiation | 1 | | |
| rsutcs | TOA outgoing clear-sky shortwave radiation | 1 | | |
| 2D atmosphere and surface variables | | | | |
| ts | surface "skin" temperature(i.e., SST for open ocean) | 1 | 1 | |
| tas | near-surface air temperature | 1 | 1 | 3 |
| tasmax | daily maximum near-surface air temperature | 1 | 1 | |
| tasmin | daily minimum near-surface air temperature | 1 | 1 | |
| uas | eastward near-surface wind | 1 | 2 | |
| vas | northward near-surface wind | 1 | 2 | |
| sfcWind | near-surface wind speed | 1 | | |
| huss | near-surface specific humidity | 1 | | |
| hurs | near-surface relative humidity | 1 | | |
| clt | total cloud fraction | 1 | 2 | |
| ps | surface air pressure | 1 | 2 | |
| psl | sea level pressure | 1 | 2 | |
| BOA fluxes | | | | |
| rlds | surface downwelling longwave radiation | 1 | 1 | |
| rlus | surface upwelling longwave radiation | 1 | 1 | |
| rsds | surface downwelling shortwave radiation | 1 | 1 | |
| rsus | surface upwelling shortwave radiation | 1 | 1 | |
| rldscs | surface downwelling clear-sky longwave radiation | 1 | 2 | |
| rsdscs | surface downwelling clear-sky shortwave radiation | 1 | 2 | |

| | | | | |
|---|---|---|---|---|
| rsuscs | surface upwelling clear-sky shortwave radiation | 1 | 2 | |
| tauu | surface downward eastward wind stress | 2 | | |
| tauv | surface downward northward wind stress | 2 | | |
| hfss | surface upward sensible heat flux | 1 | | |
| hfls | surface upward latent heat flux | 1 | | |
| pr | precipitation | 1 | 1 | 3 |
| prc | convective precipitation | 1 | 2 | |
| prsn | snowfall flux | 3 | 3 | |
| evspsbl | evaporation | 1 | | |
| Land | | | | |
| ts | skin temperature | 1 | | |
| alb | surface albedo | 1 | | |
| mrso | total soil moisture content | 1 | 3 | |
| mrfso | soil frozen water content | 1 | | |
| snd | snow depth | 1 | 3 | |
| snc | snow area fraction | 1 | | |
| snw | surface snow amount | 1 | | |
| mrro | total runoff | 1 | | |
| Sea Ice (Only for Tier-2) | | | | |
| tsice | surface temperature of sea ice | 3 | | |
| sic | sea ice area fraction | 1 | | |
| sit | sea ice thickness | 1 | | |
| snd | snow depth | 2 | | |
| hflssi | surface upward latent heat flux over sea ice | 3 | | |
| strairx | x-component of atmospheric stress on sea ice | 3 | | |
| strairy | y-component of atmospheric stress on sea ice | 3 | | |
| transix | x-component of sea ice mass transport | 3 | | |
| transiy | y-component of sea ice mass transport | 3 | | |
| 2D Ocean (Only for Tier-2; preferably on regular grid) | | | | |
| Physical variables | | | | |
| tos | sea surface temperature | 1 | | |
| hfnorth | northward ocean heat transport | 2 | | |

| | | | | |
|---|---|---|---|---|
| sltnorth | northward ocean salt transport | 2 | | |
| zos | sea surface height | 1 | | |
| zossq | square of sea surface height above geoid | 2 | | |
| zosga | global average sea level change | 2 | | |
| zossga | global average steric sea level change | 2 | | |
| zostoga | global average thermosteric sea level change | 2 | | |
| volo | sea water volume | 2 | | |
| hfds | downward heat flux at sea water surface | 1 | | |
| vsf | virtual salt flux into sea water (or equivalent fresh water flux) | 1 | | |
| Biophysical variables (Only for Tier-2; for ESMs) | | | | |
| intpp | primary organic carbon production | 2 | | |
| epc100 | downward flux of particle organic carbon | 2 | | |
| epcalc100 | downward flux of calcite | 2 | | |
| epsi100 | downward flux of particulate silica | 2 | | |
| phyc | phytoplankton carbon concentration at surface | 2 | | |
| chl | total chlorophyll mass concentration at surface | 2 | | |
| spco2 | surface aqueous partial pressure of co2 | 2 | | |
| fgco2 | gas exchange flux of co2 (positive into ocean) | 2 | | |
| 3D Atmosphere (1000, 925, 850, 700, 600, 500, 400, 300, 250, 200, 150, 100, 70, 50, 30, 20, 10 hPa) | | | | |
| ta | air temperature | 1 | | |
| ta850 | air temperature at 850 hPa | | 1 | |
| ua | eastward wind | 1 | 2 | |
| va | northward wind | 1 | 2 | |
| wap | lagrangian tendency of airpressure | 1 | 2 | |
| zg | geopotential height | 1 | | |
| zg500 | geopotential height at 500 hPa | | 1 | |
| hus | specific humidity | 1 | 2 | |
| hur | relative humidity | 1 | | |
| co2 (For ESMs) | mole fraction of $CO_2$ | 2 | | |
| 3D Ocean (Only for Tier-2; preferably on a regular grid at standard levels) | | | | |
| Physical variables | | | | |

| thetao | sea water potential temperature | 1 | | |
|---|---|---|---|---|
| so | sea water salinity | 1 | | |
| uo | sea water x velocity | 1 | | |
| vo | sea water y velocity | 1 | | |
| wo | sea water z velocity | 1 | | |
| Biophysical variables (Only for Tier-2; for ESMs) | | | | |
| dissic | Dissolved Inorganic Carbon Concentration | 2 | | |
| talk | Total Alkalinity | 2 | | |
| no3 | Dissolved Nitrate Concentration | 2 | | |
| o2 | Dissolved Oxygen Concentration | 2 | | |

**Acknowledgements**

T. Zhou acknowledges the support of International Big Science Project funded by Chinese Academy of Sciences (No. 134111KYSB20160031), and National Natural Science Foundation of China under grant Nos. 41330423 and 41125017. B.

Wu acknowledges the support of R&D Special Fund for Public Welfare Industry (meteorology) (GYHY201506012). AGT acknowledges the support of the National Centre for Atmospheric Sciences, Climate Directorate. B. Wang from IAP acknowledges the support of National Basic Research Program of China under grant No. 2014CB441302. Y. Qian's contribution is supported by the U.S. Department of Energy's Office of Science as part of the Earth System Modelling Program. The Pacific Northwest National Laboratory is operated for DOE by Battelle Memorial Institute under contract DE-

AC05-76RL01830. X. Chen acknowledges the support of China Postdoctoral Science Foundation under grant No. 2015M581152.

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

**Table 1. Description of models participating GMMIP**

| Model | Institute/Country |
|---|---|
| ACCESS | CSIRO-BOM/Australia |
| BCC-CSM2-MR | BCC/China |
| BNU-ESM | BNU/China |
| CAMS-CSM | CAMS/China |
| CanESM | CCCma/Canada |
| CAS-ESM | CAS-IAP/China |
| CESM | NCAR-COLA/USA |
| CESS-THU | THU/China |
| CMCC | CMCC/Italy |
| CNRM-CM | CNRM-CERFACS/France |
| FGOALS | IAP-LASG/China |
| FIO | FIO/China |
| GFDL | NOAA-GFDL/USA |
| GISS | NASA-GISS/USA |
| HadGEM3 | MOHC-NCAS/UK |
| IITM | IITM/India |
| IPSL-CM6 | IPSL/France |
| MIROC6-CGCM | AORI-UT-JAMSTEC-NIES/Japan |
| MPI-ESM | MPI-M/Germany |
| MRI-ESM1.x | MRI/Japan |
| NUIST-CSM | NUIST/China |

**Table 2.Experiment list of GMMIP**

| | EXP name | Integration time | Short description and purpose of the EXP design | Model type |
|---|---|---|---|---|
| Tier-1 | amip-hist | 1870-2014 | Extended AMIP run that covers 1870-2014. All natural and anthropogenic historical forcings as used in *CMIP6 Historical Simulation* will be included. AGCM resolution as *CMIP6 Historical Simulation*. The HadISST data will be used. Minimum number of integrations is 3, more realizations are encouraged. | AGCM |
| Tier-2 | hist-resIPO | 1870-2014 | Pacemaker historical run that includes all forcing as used in *CMIP6 Historical Simulation*, and the observational historical SST is restored in the tropical lobe of the IPO domain (20°S-20°N, 175°E-75°W); to understand the forcing of IPO-related tropical SST to global monsoon changes. How to restore the SST refers to the appendix. Models resolutions as *CMIP6 Historical Simulation*. The HadISST data will be used. Minimum number of integrations is 3, more realizations are encouraged. | CGCM with SST restored to the model climatology plus observational historical anomaly in the tropical lobe of IPO domain |
| | hist-resAMO | 1870-2014 | Pacemaker historical run that includes all forcing as used in *CMIP6 Historical Simulation*, and the observational historical SST is restored in the AMO domain (0°-70°N, 70°W-0°); to understand the forcing of AMO-related SST to global monsoon changes. How to restore the SST refers to the appendix. Models resolutions as *CMIP6 Historical Simulation*. The HadISST data will be used. Minimum number of integrations is 3, more realizations are encouraged. | CGCM with SST restored to the model climatology plus observational historical anomaly in the AMO domain |

|  | EXP name | Integration time | Short description and purpose of the EXP design | Model type |
|---|---|---|---|---|
| Tier-3 | amip-TIP | 1979-2014 | The topography of the TIP is modified by setting surface elevations to 500m; to understand the combined thermal and mechanical forcing of the TIP. Same model as DECK. Minimum number of integrations is 1. The topography above 500m is set to 500m in a polygon region. Coordinates of the polygon corners are as follows: longitude (from west to east ), 25 E, 40 E, 50 E, 70 E, 90 E and 180 E; latitude (from south to north), 5 N, 15 N, 20 N, 25 N, 35 N, 45 N and 75 N. The reason to remove all the topography above 500m over the Asian continent is to avoid any artificial forcings from the topography gradient when suddenly cut off at a certain height, and we also suppose the circulation response to the difference of topography between 0 to 500m can be neglected in climate models with resolutions from 100-200km. This experiment is also close to the no topography settings such as setting the topography to zero over whole Asian continent as far as possible. | AGCM |
|  | amip-TIP-nosh | 1979-2014 | Surface sensible heat released at the elevation above 500m over the TIP is not allowed to heat the atmosphere; to compare of impact of removing thermal effects. Same model as DECK. Minimum number of integrations is 1. The sensible heating is removed on the topography where is above 500m as in the same polygon region in amip-TIP; in these experiment, we have to artificially cut off the sensible heating region with a specific criterion. One practical method is set vertical temperature diffusion term to zero in the atmospheric thermodynamic equation at the bottom boundary layer (see Appendix II).There are obvious concerns over the energy conservation here, but because the suppression of heating is only in a fairly small limited area, one expects the energy balance to be compensated elsewhere. | AGCM |
|  | amip-hld | 1979-2014 | The topography of the East African Highlands in Africa and | AGCM |

Arabian Peninsula, Sierra Madre in N. America and Andes in S. America is modified by setting surface elevations to a certain height (500m) in separate experiments. Same model as DECK. Minimum number of integrations is 1. See descriptions of amip-TIP for technical details and regions as outlined in Fig. 5. The East African Highlands is in a polygon region. Coordinates of the polygon is as follows: longitude (from west to east), 27 °E and 52 °E; latitude (from south to north), 17 °S, 20 °N and 25 °N, 35 °N. Sierra Madre domain is 120 °–90 °W, 15 °–30 °N. Andes domain is 90 °–60 °W, 40 °S–10 °N.

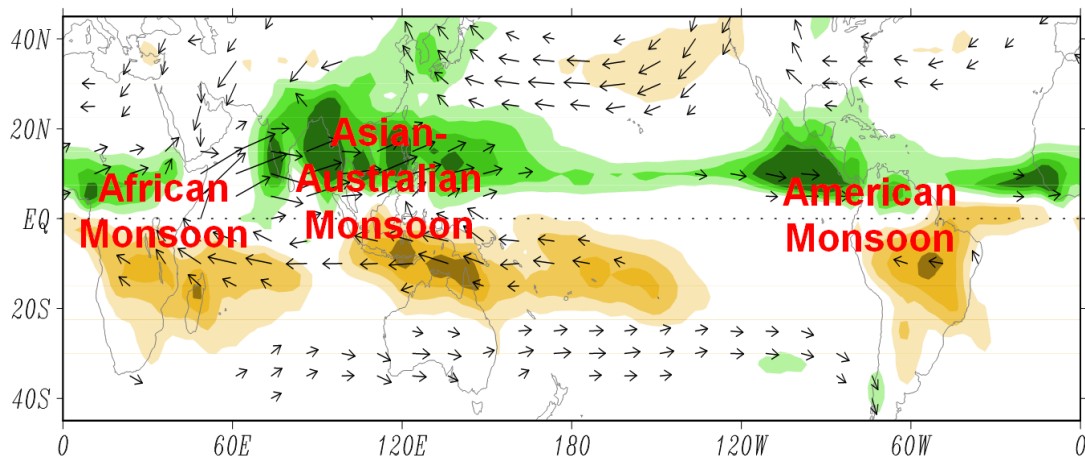

**Figure 1. Global monsoon domain and its local component, indicating by the differences of 850 hPa wind and precipitation between the June-July-August and December-January-February mean, modified from Wang and Ding (2008).**

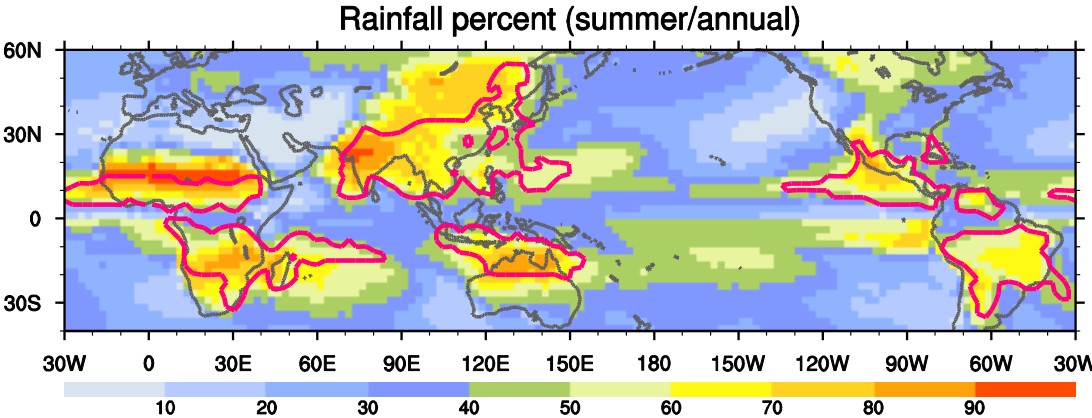

**Figure 2. Climatological percentage of summertime rainfall amount (JJAS in the Northern Hemisphere and DJFM in the Southern Hemisphere) in annual accumulation. Monsoon region is circled by red curves. GPCP data is used and the time covers 1979-2014.**

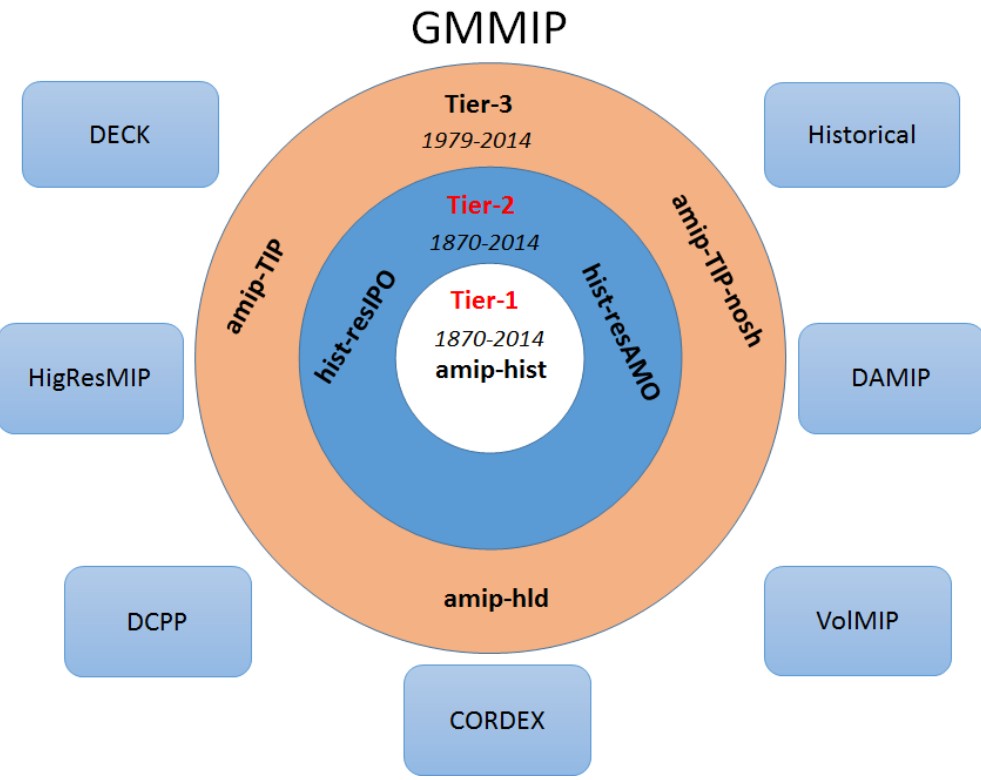

**Figure 3. Three-Tier experiments of GMMIP and its connections with DECK, Historical Simulation and endorsed MIPs.**

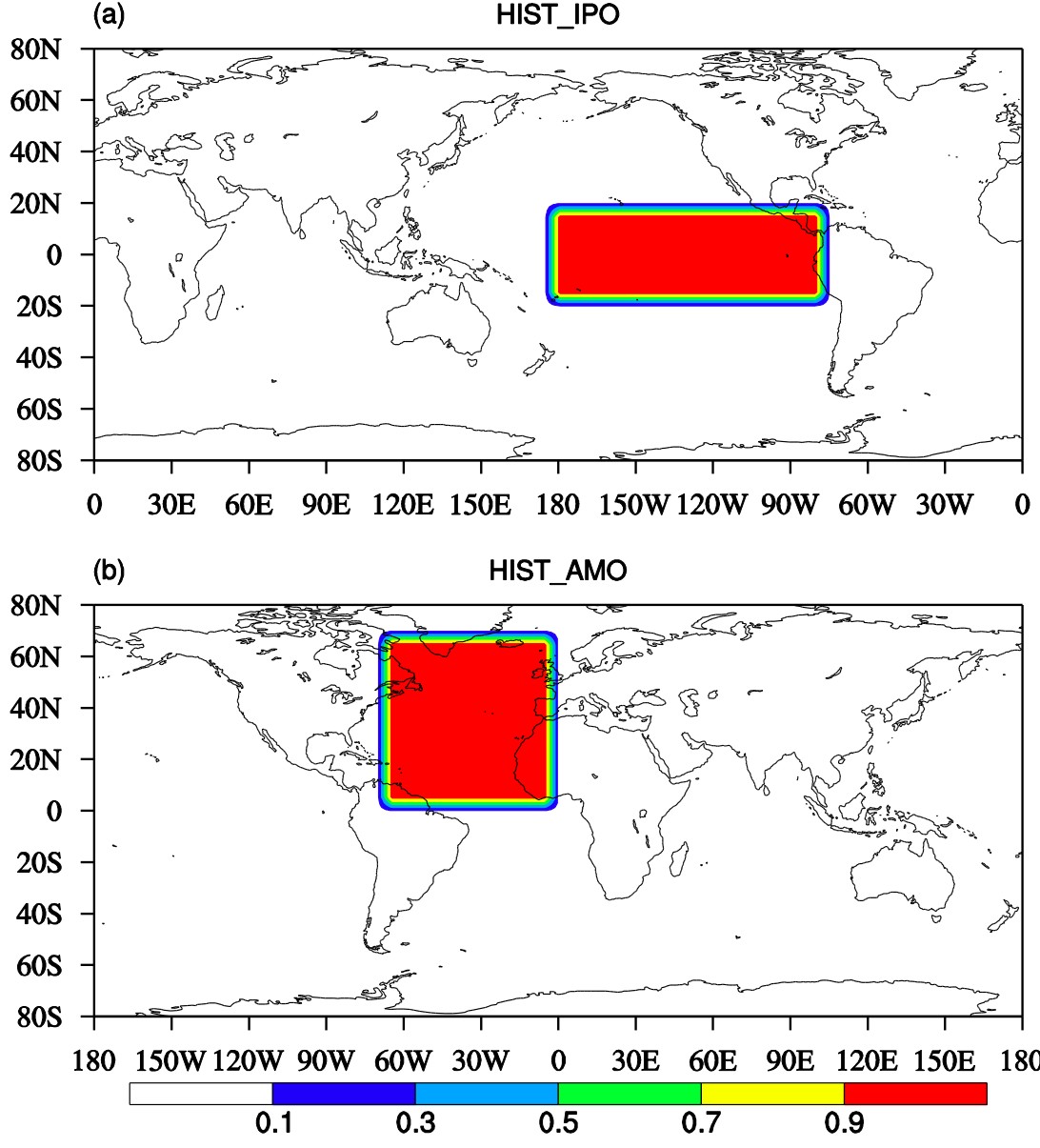

**Figure 4. The restoring regions for Tier-2 experiments HIST-IPO (a) and HIST-AMO (b).**

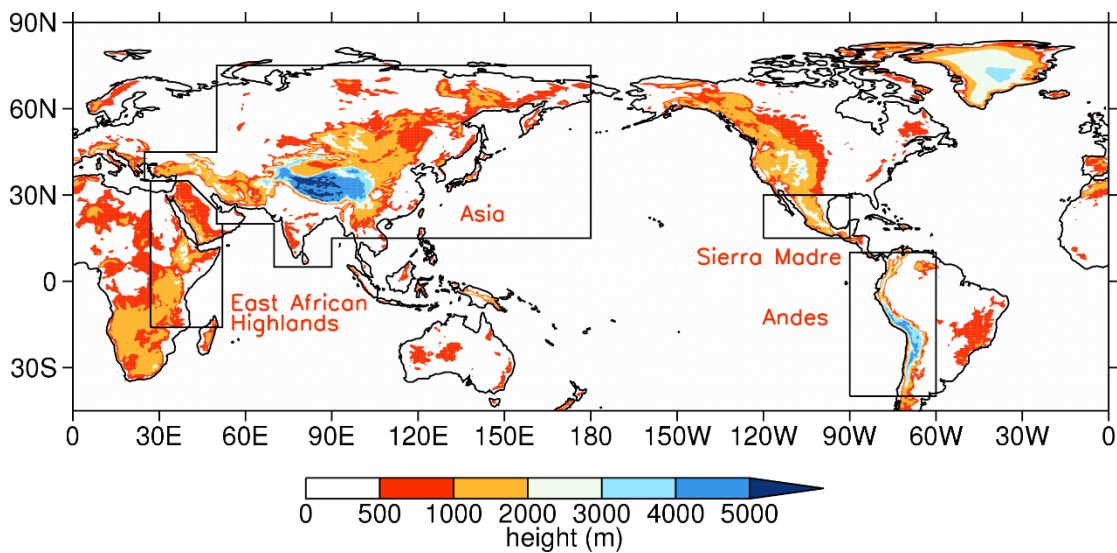

**Figure 5. The orography regions specified for the Tier-3 experiments for the Asia region (comprising the Tibetan-Iranian Plateau and Himalayas), the East African Highlands (adapted from Slingo et al., 2005), the Andes and the Sierra Madre. Within each marked region, orography would be capped at 500m height. Orographic data derived from a ~30km resolution (N512) boundary field of the Met Office HadGEM3 model.**

