# Peer review of "GMMIP (v1.0) Contribution to CMIP6: Global Monsoons Model Inter-comparison Project"

_Geoscientific Model Development, 2016_

## Short Comment (SC1) · 13 Apr 2016

Dear authors,

In agreement with the CMIP6 panel members, the Executive editors of GMD would like to establish a common naming convention for the titles of the CMIP6 experiment description papers.

The title of CMIP6 papers should include both the acronym of the MIP, and CMIP6, so that it is clear this is a CMIP6-Endorsed MIP.

Additionally, we strongly recommend to add a version number to the MIP description. The reason for the version numbers is so that the MIP protocol can be updated later, normally in a second short paper outlining the changes. See, for example:

http://www.geosci-model-dev.net/special_issue11.html,

Good formats for the title include:

'XYZMIP (v1.0) contribution to CMIP6: Name of project'

or

'Name of Project (XYZMIP v1.0) contribution to CMIP6'

If you want to include a more descriptive title, the format could be along the lines of,

'XYZMIP (v1.0) contribution to CMIP6: Name of project - descriptive title'

or

'Name of Project (XYZMIP v1.0) contribution to CMIP6: descriptive title.'

When you revise your manuscript, please correct the title of your manuscript accordingly.

Yours,

Astrid Kerkweg

---

## Short Comment (SC2) · 21 Apr 2016

Dear Astrid,

We acknowledge the request for a uniform naming convention for the manuscripts for all CMIP-endorsed MIPs associated with CMIP6. We also note the need to include a version number relating to the design.

We will make these changes during the manuscript revision process after the closure of the first round of open peer review.

On behalf of the authors,

Dr Andy Turner.

---

## Referee Comment (RC1) · W. R. Boos (Referee) · 11 May 2016

Review of "Overview of the Global Monsoons Model Inter-comparison Project (GM-MIP)" by T. Zhou et al.

Summary:

This paper presents a high-level overview of outstanding issues in monsoon variability, then proposes a series of climate model integrations that might be used to better understand the causes of monsoon variability. The introduction is well-written and concise, and does a particularly nice job of quickly summarizing what is known and not known about the coupling between regional and global-scale variations in monsoon circulations. The idea of a model intercomparison focusing on monsoons is well-motivated and compelling, and I am sure that new understanding will be generated by this work.

But some aspects of the experimental design should be clarified and perhaps modified. The "orographic perturbation" experiments do not seem designed to address scientific questions for which there remains considerable uncertainty, and there is some lack of clarity in the associated methodology. The possibility that model bias may interfere with the ability to draw conclusions should be given more consideration. I list more details on these major issues below, along with some minor technical details.

Major scientific issues:

1. Most of the proposed "orographic perturbation" experiments are not appropriately designed to test any hypotheses for which there exists considerable uncertainty. There are several key issues here:

a. It is widely agreed that eliminating all elevated topography from climate models results in a dramatic weakening and southward shift of South Asian monsoon rainfall; this was shown in Hahn and Manabe (1975), Prell and Kutzbach (1992), Boos and Kuang (2010), Wu et al. (2012), and others, with no disagreement amongst those papers. So it seems strange to devote simulations by such a large number of modeling groups to verifying this well-accepted result.

b. The manuscript overstates the controversy concerning ways in which Asian orography affects the monsoon. I would agree that there is a widespread belief that controversy exists, but if one actually reads the recent literature one will find little actual disagreement. Wu et al. (2012) clearly state that elevated orographic heating is primarily important for a "northern branch" of the South Asian monsoon that exists north of 20N and lies "along the southern margin of the Iranian Plateau-Tibetan Plateau in the subtropics." That view is very consistent with Boos and Kuang (2010), who showed that Tibetan Plateau surface enthalpy fluxes indeed produced a large fraction of summer rainfall along the plateau's southern margin, but made negligible contribution to the interhemispheric monsoon circulation and the main rainfall maxima, both of which lie south of 20N. Boos (2013, CLIVAR Exchanges) reviewed the agreement between Wu

et al. 2012 and Boos and Kuang 2010, and discussed the lack of disagreement in recent literature concerning the influence of topography on the South Asian monsoon. So while it would be interesting to see results from the proposed orographic perturbation experiments, I think the authors should seriously consider whether it is desirable to use such a large amount of modeling and computational resources to examine something that is not fundamentally controversial when one reads the literature closely.

c. Turning off sensible heat fluxes from all Asian topography higher than 500 m in the proposed "TIP" domain amounts to imposing a huge negative heat sink over roughly half of the Asian continent. The authors propose to suppress sensible heat fluxes from most of the red and orange regions in the "Asia" box in Fig. 5, which includes parts of continental India as well as much of China and Mongolia — regions not thought to be involved in "elevated heating" when it is discussed in the monsoon literature. In other words, it would be surprising if the monsoon did not weaken when surface sensible heat fluxes were suppressed over one-third to one-half of Asia, whether or not that terrain was elevated! These experiments thus don't clearly test the idea that elevated heating from Tibet or from the slopes of the Himalaya are a key forcing for the South Asian monsoon (and as stated above, both Wu et al. 2012 and Boos and Kuang 2010 already agree that elevated heating from those regions forces precipitation along the Himalayas but not the interhemispheric South Asian monsoon circulation). Finally, modern theory for tropical atmospheric dynamics places surface latent heat fluxes on the same footing as surface sensible heat fluxes in their influence on large-scale flow (e.g. see theories for convective quasi-equilbirium, reviewed by Emanuel et al. 1994 QJRMS, or theories for the energy flux equator discussed by Kang et al. 2008, J. Climate p. 3521), so it is unclear why there should be a special emphasis on surface sensible heat fluxes. I thus suggest the authors reconsider the design of the TIP-NSH experiment.

d. The methodology for eliminating the surface sensible heat flux in the orographic perturbation experiments is unclear and may lead to different approaches being taken

by different modeling groups. The manuscript states that, as in Wu et al. (2012), surface sensible heating will be suppressed by setting "the vertical diffusive heating term in the atmospheric thermodynamic equation" to zero. But does this mean that heat will accumulate just above the surface and will not diffuse upward through the boundary layer, so that the column will eventually become unstable to dry convection or to grid-scale overturning? And how exactly does suppressing this vertical diffusion alter the land surface energy budget ... e.g. will land surface temperatures and longwave emission become very high because heat cannot diffuse away from the land surface? Participating models may have dramatically different methods of parameterizing the subgrid scale vertical redistribution of surface sensible heat fluxes. If one wanted to suppress surface heat fluxes (which is debatable, see previous point) it would seem better to prescribe a heat sink in the bottom layer of the atmosphere that is exactly equal to the surface sensible heat flux at that time step. Then the net land surface energy budget will not be directly altered, the surface sensible heat flux will not heat the atmosphere, and one does not need to worry about the various ways in which different models represent vertical diffusion.

2. This manuscript seems to assume that model bias will not compromise the ability of the proposed experiments to provide insight on the cause of monsoon variability. For example, the authors state at top of p. 6 that comparing prescribed SST integrations with fully coupled integrations will allow the authors "to determine the importance of SST variability to long and short-term trends in the monsoons." But later they state that "simulations with specified SST generally have low skill in simulating the interannual variation of the summer precipitation over global monsoon domains". So it is very possible that the specified SST integrations will have such large bias that it will not be possible to use them to understand long- and short-term trends. This problem is difficult, at best, to fix, but I would have at least liked to see more acknowledgment of this problem and more attempts to gauge model skill through comparison with observations. For example, the authors state that comparison of pre-industrial control simulations with the Tier-2 experiments will "allow us to determine which parts of ap-

parent decadal variations in the monsoons are caused by underlying SST, and which are forced solely from externally driven sources, such as volcanic emissions." But what if all of the models have a strongly biased response to volcanic emissions? Some users of the GMMIP archive might compare with observations and stratify models by their skill in simulating, e.g., the response to Pinatubo, but this cannot be assumed — there are numerous examples of model intercomparisons in which every model in an ensemble is treated equally. The bottom line is that I suggest more discussion of the possibility that model bias will make it difficult to draw conclusions about causation, and more concrete proposals for how to deal with this bias if it is found to exist. Otherwise one runs the risk of gaining little new understanding from the proposed large amounts of simulation.

Minor technical issues:

3. After the introduction, the manuscript quickly becomes somewhat difficult to read for those who are not deeply familiar with the CMIP terminology. This could be easily remedied by clearly explaining the meaning of various terms when they are first introduced. E.g. what are the "DECK" experiments? What is a "pacemaker" experiment? It is possible for the reader to figure out what is meant by a pacemaker experiment, but a clearer statement and references to literature discussing the history and caveats of pacemaker experiments would be very helpful. On p. 4, line 31 the terms "Tier-1" and "Tier-3" are used without being previously defined, and I was confused about what these terms meant until they were defined a full page later.

4. Unclear what is meant by "model climatology" on p. 6, line 8. Is this a cyclic seasonal cycle of daily resolution, or the full, interannually varying daily time series of SST from the coupled CMIP6 integration?

5. Equation (1) is introduced in method (b), but it also defines the "constructed SST" introduced in (a), with the linear decay of the relaxation time in the buffer zone already "built in". My point is that it would seem more clear to introduce equation (1) in method

(a).

6. Page 12, line 9: isn't 50 m a very deep mixed layer depth for the East Pacific, which is the main region of interest for the "IPO" pacemaker experiment? This could result in a factor of 2 or more difference in the effective restoring times for SST in the IPO and AMO pacemaker experiments. Would at least be nice to see some mention of why it's acceptable to use a 50 m mixed layer depth in the East Pacific.

7. The box marked around the "Asia" domain in Fig. 5 does not agree with the coordinates given in Table 2. Should there be agreement? If not, what do the boxes in Fig. 5 represent?

Signed, William Boos

---

## Referee Comment (RC2) · Anonymous Referee #2 · 13 May 2016

This is an overview of one of satellite MIPs under CMIP6, which is already endorsed by the CMIP6 panel. Therefore only minor comments are given here.

Minor comments: (1) page 2, line 7: The East Asian monsoon is controlled by zonal temperature and pressure gradient. Therefore, "meridional temperature and pressure gradients" should be replaced with "temperature and pressure gradients" without "meridional". (2) page 4: Four primary scientific questions are raised here, but how predictability of monsoons can be solved by GMMIP is unclear. Delete this question or include one sub-section regarding this in Section 5. (3) page 5, line 8: regional climate information is not a part of WCRP Grand Challenges (unfortunately). (4) page 8, Section 5: How is CORDEX data planned to use? (5) page 9, line 29: A maximum width of the Meiyu/Baiu rain band is about 200 km in a climatological time averaging, but it consists of meso-scale cloud clusters. This is why high-resolution modeling is needed.

(6) page 11: In the pacemaker experiments, SST is restored to daily climatological SST. On the other hand, in the AMIP experiment, the Taylor-corrected monthly mean SST is used after interpolation into daily values. Therefore temporal behavior of SST is different between the AMIP and the pacemaker experiments. Doesn't this matter?

———————————————————

---

## Short Comment (SC3) · 24 May 2016

The manuscript deals with the description of the GMMIP experiments in the framework of the next CMIP6 effort. The Introduction is a nice overview of the main issues of monsoon variability and simulation, including still unsolved shortcoming in monsoon modelling that should benefit from the experiments and the comparison proposed. Overall the manuscript is well structured, however I have few general comments and some technical corrections as listed below.

**General comments**:

1 - Why the term "global monsoons" is plural? The global monsoon represents the global hydrological cycle and it is very important/interesting to have metrics to consider it as a single phenomenon. Nevertheless it is composed by the regional monsoons. I

think it is important to stress on the manuscript the need to have both, as this would help merging the contribution from the different communities dedicated to the regional monsoons (actually this is done in some parts, I would check it to be consistent in the whole manuscript)

2 - In the Introduction the issue of the recent observed decrease in precipitation over India and the tendency of the coupled models to have increased precipitation when the atmospheric $CO_2$ increases should be discussed (i.e. issues of thermodynamical versus dynamical changes in precipitation as discussed in Cherchi et al., 2011 and in Endo and Kitoh, 2014 for the different monsoon regions) - see references: Cherchi et al. (2011) Clim Dyn 37 83-101 doi:10.1007/s00382-010-0801-7; Endo and Kitoh (2014) GRL 41 1704-1710 doi:10.1002/2013GL059158.

3 - Table 1: a useful information that should be added in this table and that should be mentioned in paragraphs 4.2 4.3 4.4 and 4.5 is the models involved in GMMIP that will be also involved in the other respective MIPs. This would help to know how many models (i.e. how large will be the sample) could be included in the comparison

4 - You should specify if you have specific requirements for the variables (and respective time-frequency) that should be saved as output from the GMMIP experiments (they should be listed in the manuscript)

5 - You should specify what specific criterion should be used for the TIP-NSH experiment (tier-3) to cut off the sensible heating from the selected regions

**Some technical corrections**:

Page 1, line 20: change "during" with "in"

Page 1, line 23: remove the comma after the word "DECK"

Page 1, line 27: I would use "benefit monsoons prediction .." instead of "benefit monsoon prediction .."

Page 2, lines 15-18: I think that in the Introduction the issue of internal feedback should be separated from that of external driven processes, and discussed in more detail

Page 6, sections 3.2 and 3.3: what are the initial conditions for these experiments? how long are these experiments? I would suggest including these information also in the text not only in the table

Page 6, line 15: I would insert "coupled" between "historical" and "climate"

Page 6, lines 23-24: I think it is better to consider the tier-3 as a perturbation of the Tier-1 rather to the DECK (it is the same, it is just a matter of flow of the description)

Page 7, line 5: what do you mean by "standard CMIP6 horizontal and vertical resolutions"

Page 7, section 4.1: the chain of comparisons between different experiments is a bit confusing. Consider rewriting the paragraph. More for the comparison of Tier-2 experiments with pre-industrial and historical simulations, please consider that in the former (tier-2 experiments) you have prescribed SST in selected regions but you have also the contribution of anthropogenic GHG and aerosols

Page 7, lines 28-29: not clear, please rewrite. Why high resolution in the mid-latitudes?

Page 7-8, section 4.3: It should be mentioned in the manuscript that in HighResMIP the SST used to build the AMIP experiments will be used as daily mean, differently from the other AMIP protocol. This should be considered also for the kind of comparison that would result. Also in HighResMIP the aerosols would be sort of prescribed (mandatory use of MPI simple plume module for anthropogenic aerosols). This should also be mentioned and discussed in terms of possible comparisons with GMMIP experiments

Page 8, lines 24-26: this could also be a hard comparison because of the specificity of the HighResMIP experiments as mentioned in the comment just above. You should mention what kind of specific metrics/analysis could be used/you have in mind for this comparison?

Page 9, line 7: "ACGM" should be "AGCM"

Page 9, lines 7-9: both 20CR and ERA20C are global atmospheric reanalyses that assimilate only the surface pressure (and the SST are prescribed)

Page 9, line 7-8: you should include references for 20CR and ERA20C

Page 9, line 15: "global monsoon" instead of "global monsoons"

Page 11: "Data availability" should be an appendix, I guess (see also general comment above for specific requirements on variables and related time frequency)

Page 11, line 25: insert "coupled" between "historical" and "simulation"

---

## Short Comment (SC4) · 7 Jun 2016

The CMIP Panel is undertaking a review of the CMIP6 GMD special issue papers to ensure a level of consistency among the invited contributions, and to consider whether the co-chairs adequately addressed the key questions outlined in our request to submit a paper. We very much welcome the important contribution from GMMIP to the CMIP6 special issue, below are a few comments:

- Thank you for responding to our request (through the editor) promising to revise your title consistent with others in this special issue.

- It would be helpful to replace "other MIPs" (and similar constructs) in some places with "endorsed MIPs" to make clear the cohesiveness of the CMIP6 set.

- There is essentially no discussion of any special data required for your analyses.

- Perhaps there are no special needs, but it would be good to at least include some discussion of the priority 1 data request. In particular, if data with temporal resolution greater than a month are needed, then indicate this, especially if 3-d (spatially) fields are required. Without emphasizing this, some groups may not realize the importance of saving the needed fields.

With many thanks for your ongoing efforts in the CMIP6 process.

Karl Taylor, reviewing on behalf of the CMIP Panel

---

## Author Comment (AC1) · 21 Jul 2016

Dear Astrid,

Thanks for your suggestion. We have revised the title as "GMMIP (v1.0) Contribution to CMIP6: Overview of the Global Monsoons Model Inter-comparison Project".

On behalf of the authors,

Dr Xiaolong Chen

---

## Author Comment (AC4) · 21 Jul 2016

Dear Karl,

Thank you for your constructive suggestions. For reading easily, we copied your comments in italic.

*- Thank you for responding to our request (through the editor) promising to revise your title consistent with others in this special issue.*
*- It would be helpful to replace "other MIPs" (and similar constructs) in some places with "endorsed MIPs" to make clear the cohesiveness of the CMIP6 set.*

**Response:**
Revised as suggested.

[Figure]

*- There is essentially no discussion of any special data required for your analyses.*
*- Perhaps there are no special needs, but it would be good to at least include some discussion of the priority 1 data request. In particular, if data with temporal resolution greater than a month are needed, then indicate this, especially if 3-d (spatially) fields are required. Without emphasizing this, some groups may not realize the importance of saving the needed fields.*

**Response:**
We now have added Appendix II to clarify what kinds of data are required **(P13-16)**.
* * *

---

## Author Response (AR1)

**In the following, the texts with italic font are the reviewer's original comments, and the texts with normal font are authors' response. The revised parts of the manuscript are marked by red.**

Review comments:

**Reviewer #1: A. Kerkweg**

*Dear authors,*

*In agreement with the CMIP6 panel members, the Executive editors of GMD would like to establish a common naming convention for the titles of the CMIP6 experiment description papers.*

*The title of CMIP6 papers should include both the acronym of the MIP, and CMIP6, so that it is clear this is a CMIP6-Endorsed MIP.*

*Additionally, we strongly recommend adding a version number to the MIP description. The reason for the version numbers is so that the MIP protocol can be updated later, normally in a second short paper outlining the changes. See, for example: Printer-friendly version Discussion paper*

*http://www.geosci-model-dev.net/special_issue11.html,*

*Good formats for the title include:*

*'XYZMIP (v1.0) contribution to CMIP6: Name of project' or 'Name of Project (XYZMIP v1.0) contribution to CMIP6'*

*If you want to include a more descriptive title, the format could be along the lines of, 'XYZMIP (v1.0) contribution to CMIP6: Name of project - descriptive title' or 'Name of Project (XYZMIP v1.0) contribution to CMIP6: descriptive title.'*

*When you revise your manuscript, please correct the title of your manuscript accordingly.*

*Yours,*

*Astrid Kerkweg*

**Response:**

Thanks for your suggestion. We have revised the title as "GMMIP (v1.0) Contribution to CMIP6: Overview of the Global Monsoons Model Inter-comparison Project" **(P1, L1-2)**

**Reviewer #2: W. R. Boos (Referee)**

*This paper presents a high-level overview of outstanding issues in monsoon variability, then proposes a series of climate model integrations that might be used to better understand the causes of monsoon variability. The introduction is well-written and concise, and does a particularly nice job of quickly summarizing what is known and not known about the coupling between regional and global-scale variations in monsoon circulations. The idea of a model intercomparison focusing on monsoons is well-motivated and compelling, and I am sure that new understanding will be generated by this work.*

*But some aspects of the experimental design should be clarified and perhaps modified. The "orographic perturbation" experiments do not seem designed to address scientific questions for which there remains considerable uncertainty, and there is some lack of clarity in the associated methodology. The possibility that model bias may interfere with the ability to draw conclusions should be given more consideration. I list more details on these major issues below, along with some minor technical details.*

*Major scientific issues:*

*1. Most of the proposed "orographic perturbation" experiments are not appropriately designed to test any hypotheses for which there exists considerable uncertainty. There are several key issues here:*

*a. It is widely agreed that eliminating all elevated topography from climate models results in a dramatic weakening and southward shift of South Asian monsoon rainfall; this was shown in Hahn and Manabe (1975), Prell and Kutzbach (1992), Boos and Kuang (2010), Wu et al. (2012), and others, with no disagreement amongst those papers. So it seems strange to devote simulations by such a large number of modeling groups to verifying this well-accepted result.*

**Response:**

Thanks for the comments. While the large-scale response of eliminating all elevated topography from climate models is almost the same among the published papers, regional scale features are different. The aim of the "orographic perturbation"

experiment is to quantitatively understand the regional response to the orographic perturbation from both the thermal and dynamical perspectives. Because the model dynamics and physics are different among the CMIP6 models, the response in each model may be different across temporal and spatial scales. The results will be very helpful to quantitatively understand the topography effect on the atmosphere and associated physical processes, such as the distribution, intensity, and frequency changes in the precipitation over wide monsoon regions. In addition, the"orographic perturbation" experiments are listed as Tier-3 experiments, viz. the lowest priority, although we wish a large number of modeling groups would do the experiments, the experiments probably will be done only in several modeling centers majored in monsoon research.

*b. The manuscript overstates the controversy concerning ways in which Asian orography affects the monsoon. I would agree that there is a widespread belief that controversy exists, but if one actually reads the recent literature one will find little actual disagreement. Wu et al. (2012) clearly state that elevated orographic heating is primarily important for a "northern branch" of the South Asian monsoon that exists north of 20N and lies "along the southern margin of the Iranian Plateau-Tibetan Plateau in the subtropics." That view is very consistent with Boos and Kuang (2010), who showed that Tibetan Plateau surface enthalpy fluxes indeed produced a large fraction of summer rainfall along the plateau's southern margin, but made negligible contribution to the interhemispheric monsoon circulation and the main rainfall maxima, both of which lie south of 20N. Boos (2013, CLIVAR Exchanges) reviewed the agreement between Wu et al. 2012 and Boos and Kuang 2010, and discussed the lack of disagreement in recent literature concerning the influence of topography on the South Asian monsoon. So while it would be interesting to see results from the proposed orographic perturbation experiments, I think the authors should seriously consider whether it is desirable to use such a large amount of modeling and computational resources to examine something that is not fundamentally controversial when one reads the literature closely.*

**Response:**

Thanks. In the revised version, we replaced the statement of "the relative roles of the two effects remain controversial" with "the relative roles of the two effects deserve further investigation" **(P4, L2)**. The statement in section 5.4 of "remains debatable" is replaced with "needs further study" **(P10, L25)**. In addition, the model resolutions in most old GCMs (100km or so) are not high enough to resolve the complex topography over south slope of TP and we hope the higher resolution models of CMIP6 could better resolve these effects. Recent progress indicates that the surface entropy over northern India is quite sensitive to the large-scale thermal forcing of TP and cannot be solely attributed to the barrier effect of TP (Wu et al., 2015, Climate Dynamics; He et al., 2015, Scientific Reports). Since these results may be model-dependent, we hope other modeling centers can also do the experiments. In addition, this experiment is listed as Tier-3 experiment (viz. low priority) and honestly we expect only a few modeling centers that have specific interest in the monsoon to do the experiment.

*c. Turning off sensible heat fluxes from all Asian topography higher than 500 m in the proposed "TIP" domain amounts to imposing a huge negative heat sink over roughly half of the Asian continent. The authors propose to suppress sensible heat fluxes from most of the red and orange regions in the "Asia" box in Fig. 5, which includes parts of continental India as well as much of China and Mongolia — regions not thought to be involved in "elevated heating" when it is discussed in the monsoon literature.*

*In other words, it would be surprising if the monsoon did not weaken when surface sensible heat fluxes were suppressed over one-third to one-half of Asia, whether or not that terrain was elevated! These experiments thus don't clearly test the idea that elevated heating from Tibet or from the slopes of the Himalaya are a key forcing for the South Asian monsoon (and as stated above, both Wu et al. 2012 and Boos and Kuang 2010 already agree that elevated heating from those regions forces precipitation along the Himalayas but not the interhemispheric South Asian monsoon*

*circulation). Finally, modern theory for tropical atmospheric dynamics places surface latent heat fluxes on the same footing as surface sensible heat fluxes in their influence on large-scale flow (e.g. see theories for convective quasi-equilbirium, reviewed by Emanuel et al. 1994 QJRMS, or theories for the energy flux equator discussed by Kang et al. 2008, J. Climate p. 3521), so it is unclear why there should be a special emphasis on surface sensible heat fluxes. I thus suggest the authors reconsider the design of the TIP-NSH experiment.*

**Response:**

We agree that the latent heat fluxes are very important over the tropical oceans to produce low level instability for moist convection. However over the land in Asia, the link of local evaporation and precipitation is relatively weak (He et al., 2015, Scientific Reports), and the sensible heat flux is the major term which causes the PV anomaly at the surface to draw water vapor from the Indian Ocean. Meanwhile the latent heat flux (evaporation) is also affected by the SH. Therefore the SH is regarded as the main driver of the behavior of the low level atmosphere and possibly also the upper troposphere and lower stratosphere (Wu et al., 2016, *Science China Earth Sciences*). The importance of elevated heating has been emphasized by He (2016, *Theor. Appl. Climatol*.), and the responses can be obtained from the differences between AMIP-TIP-nosh and AMIP (which is the control run). Again, to examine whether the responses are model dependent, we hope other modeling centers will do the experiment. This experiment is listed as Tier-3.

*d. The methodology for eliminating the surface sensible heat flux in the orographic perturbation experiments is unclear and may lead to different approaches being taken by different modeling groups. The manuscript states that, as in Wu et al. (2012), surface sensible heating will be suppressed by setting "the vertical diffusive heating term in the atmospheric thermodynamic equation" to zero. But does this mean that heat will accumulate just above the surface and will not diffuse upward through the boundary layer, so that the column will eventually become unstable to dry convection or to grid-scale overturning? And how exactly does suppressing this*

*vertical diffusion alter the land surface energy budget ... e.g. will land surface temperatures and longwave emission become very high because heat cannot diffuse away from the land surface? Participating models may have dramatically different methods of parameterizing the subgrid scale vertical redistribution of surface sensible heat fluxes. If one wanted to suppress surface heat fluxes (which is debatable, see previous point) it would seem better to prescribe a heat sink in the bottom layer of the atmosphere that is exactly equal to the surface sensible heat flux at that time step. Then the net land surface energy budget will not be directly altered, the surface sensible heat flux will not heat the atmosphere, and one does not need to worry about the various ways in which different models represent vertical diffusion.*

**Response:**

The sensible heating (vertical heat diffusion in atmosphere) is set to zero for each step in the atmospheric model, and will not be accumulated at the model surface. The sensible heat flux in the atmospheric model is zero (Fig. R1a), while the sensible heat flux in the land model continues to be updated (Fig. R1b), and the procedure will not affect the land surface energy conservation (Fig. R1c), and the net surface radiation balance on the surface of the atmosphere model is reasonable (Fig. R1d).

[Figure]

Fig. R1. The evolution (the abscissa is in units of months from the startup run of the model) of various variables anomaly in the IAP/LASG atmospheric model (SAMIL) and land model (CLM) in the amip-TIP-nosh experiment respectively: (a) Sensible heat flux in atmosphere model, (b) sensible heat flux in land model, (c), the error in energy conservation in land model, and (d) the net surface radiation in atmosphere model.

*2. This manuscript seems to assume that model bias will not compromise the ability of the proposed experiments to provide insight on the cause of monsoon variability. For example, the authors state at top of p. 6 that comparing prescribed SST integrations with fully coupled integrations will allow the authors "to determine the importance of SST variability to long and short-term trends in the monsoons." But later they state that "simulations with specified SST generally have low skill in simulating the interannual variation of the summer precipitation over global monsoon domains". So it is very possible that the specified SST integrations will have such large bias that it will not be possible to use them to understand long- and short-term trends. This problem is difficult, at best, to fix, but I would have at least liked to see more acknowledgment of this problem and more attempts to gauge model skill through comparison with observations. For example, the authors state that comparison of pre-industrial control simulations with the Tier-2 experiments will "allow us to determine which parts of apparent decadal variations in the monsoons are caused by underlying SST, and which are forced solely from externally driven sources, such as volcanic emissions." But what if all of the models have a strongly biased response to volcanic emissions? Some users of the GMMIP archive might compare with observations and stratify models by their skill in simulating, e.g., the response to Pinatubo, but this cannot be assumed — there are numerous examples of model intercomparisons in which every model in an ensemble is treated equally. The bottom line is that I suggest more discussion of the possibility that model bias will make it difficult to draw conclusions about causation, and more concrete proposals*

*for how to deal with this bias if it is found to exist. Otherwise one runs the risk of gaining little new understanding from the proposed large amounts of simulation.*

**Response:**

Thanks for comments. Yes, climate models have been showing and will continue to show bias in many aspects. We have to balance the needs of scientific research and the performances of the current state of the art climate models. A multi-model intercomparison approach is a useful way to provide insights for reducing the uncertainty due to model bias. This is the reason why MIPs for CMIP6 are needed. As suggested, the impact of model bias on the conclusion should be discussed. We have added a paragraph in the revised manuscript:

"Current state-of-the-art climate models still show bias in the simulation of monsoon (Sperber et al. 2013). We acknowledge that attention should be paid to the model bias in the analysis of model outputs, although multi-model ensemble/intercomparison approach is a useful way to better quantify the uncertainty related to model bias" **(P10, L2-4)**.

In addition, the analysis of GMMIP will focus on both monsoon circulation and monsoon precipitation. Although SST-driven AGCM simulations generally have low skill in the simulation of monsoon precipitation over the Asian-Australian monsoon domain due to the neglect of air-sea coupling and model bias, the large scale monsoon circulation changes have significant skill at both interannual and inter-decadal time scales. This has been demonstrated in many published papers. Thus at the top of p. 6 of the original manuscript, we revised the statement: "to determine the importance of SST variability to long and short-term trends in the monsoon circulations and the associated precipitation" **(P6, L4-5)**.

*Minor technical issues:*

*3. After the introduction, the manuscript quickly becomes somewhat difficult to read for those who are not deeply familiar with the CMIP terminology. This could be easily remedied by clearly explaining the meaning of various terms when they are first introduced. E.g. what are the "DECK" experiments? What is a "pacemaker"*

*experiment? It is possible for the reader to figure out what is meant by a pacemaker experiment, but a clearer statement and references to literature discussing the history and caveats of pacemaker experiments would be very helpful. On p. 4, line 31 the terms "Tier-1" and "Tier-3" are used without being previously defined, and I was confused about what these terms meant until they were defined a full page later.*

**Response:**

Thanks. We have revised as suggested. In addition, a paper of CMIP6 design which clearly described the CMIP6 core experiments such as DECK has been cited **(P4, L29)**:

Eyring, V., Bony, S., Meehl, G. A., Senior, C. A., Stevens, B., Stouffer, R. J., and Taylor, K. E.: Overview of the Coupled Model Intercomparison Project Phase 6 (CMIP6) experimental design and organization, Geosci. Model Dev., 9, 1937-1958, doi:10.5194/gmd-9-1937-2016, 2016.

*4. Unclear what is meant by "model climatology" on p. 6, line 8. Is this a cyclic seasonal cycle of daily resolution, or the full, interannually varying daily time series of SST from the coupled CMIP6 integration?*

**Response:**

It is the former. We have clarified this in Appendix I **(P12, L12)**.

*5. Equation (1) is introduced in method (b), but it also defines the "constructed SST" introduced in (a), with the linear decay of the relaxation time in the buffer zone already "built in". My point is that it would seem more clear to introduce equation (1) in method (a).*

**Response:**

We added an equation for method (a) to make the difference between (a) and (b) clearer **(P12, L15)**.

*6. Page 12, line 9: isn't 50 m a very deep mixed layer depth for the East Pacific, which is the main region of interest for the "IPO" pacemaker experiment? This could*

*result in a factor of 2 or more difference in the effective restoring times for SST in the IPO and AMO pacemaker experiments. Would at least be nice to see some mention of why it's acceptable to use a 50 m mixed layer depth in the East Pacific.*

**Response:**

The choice of 50m for hist-resIPO is based on Kosaka and Xie (2013, *Nature*). For the hist-resAMO experiment, we use a restoring time of 2 months following the DCPP Component C experiments (Boer et al., 2016). For hist-resIPO, 10m typical mixed layer and 10-day restoring time are used so the restoring intensity is comparable between hist-resIPO and hist-resAMO **(P13, L1-6)**

*7. The box marked around the "Asia" domain in Fig. 5 does not agree with the coordinates given in Table 2. Should there be agreement? If not, what do the boxes in Fig. 5 represent?*

**Response:**

The domain description has been revised. Now the text is consistent with the figure **(P25)**.

**Reviewer #3: Anonymous Referee**

*This is an overview of one of satellite MIPs under CMIP6, which is already endorsed by the CMIP6 panel. Therefore only minor comments are given here.*

*Minor comments:*

*(1) page 2, line 7: The East Asian monsoon is controlled by zonal temperature and pressure gradient. Therefore, "meridional temperature and pressure gradients" should be replaced with "temperature and pressure gradients" without "meridional".*

**Response:**

Done **(P2, L8)**.

*(2) page 4: Four primary scientific questions are raised here, but how predictability of monsoons can be solved by GMMIP is unclear. Delete this question or include one sub-section regarding this in Section 5.*

**Response:**

The term "predictability" is replaced with "reproducibility" in the revision **(P4, L16)**. The interannual variability of monsoons simulated by stand-alone AGCMs will be compared to the results of fully coupled models. The impact of air-sea interaction in the reproducibility of interannual monsoon variation will be addressed.

*(3) page 5, line 8: regional climate information is not a part of WCRP Grand Challenges (unfortunately).*

**Response:**

We deleted this part. In the draft version of WCRP grand challenge documentation, the regional climate information was among the list.

*(4) page 8, Section (5): How is CORDEX data planned to use?*

**Response:**

This has been clarified in the revision **(P9, L7-11)**.

    "In the core framework of CORDEX phase 2 (CORDEX2 hereafter), a core set of regional climate models (RCMs) downscales a core set of GCMs over all or most

CORDEX domains at 10-20 km resolutions (Gutowski Jr., et al. 2016). The comparisons of CORDEX2 historical climate downscaling with the driving GCMs historical simulations, will give insight into the importance of model resolution and the added value of RCMs in the simulation of climatology and variability of global monsoon, especially the global land monsoon."

*(5) page 9, line 29: A maximum width of the Meiyu/Baiu rain band is about 200 km in a climatological time averaging, but it consists of meso-scale cloud clusters. This is why high-resolution modeling is needed.*

**Response:**

Thanks. We have revised the statement as suggested (**P10, L18-19**).

*(6) page 11: In the pacemaker experiments, SST is restored to daily climatological SST. On the other hand, in the AMIP experiment, the Taylor-corrected monthly mean SST is used after interpolation into daily values. Therefore temporal behavior of SST is different between the AMIP and the pacemaker experiments. Doesn't this matter?*

**Response:**

In the pacemaker experiments, the SST is restored to a constructed SST which is climatological model SST plus observed anomalies to reduce model drift. We suggest using the same SST data as in AMIP experiments to calculate the observational anomalies. So variability at all the time scales is the same between these two types of experiments (**P12, Appendix I**).

**Reviewer #4: A. Cherchi**

*The manuscript deals with the description of the GMMIP experiments in the framework of the next CMIP6 effort. The Introduction is a nice overview of the main issues of monsoon variability and simulation, including still unsolved shortcoming in monsoon modelling that should benefit from the experiments and the comparison proposed. Overall the manuscript is well structured, however I have few general comments and some technical corrections as listed below.*

*General comments:*

*1 - Why the term "global monsoons" is plural? The global monsoon represents the global hydrological cycle and it is very important/interesting to have metrics to consider it as a single phenomenon. Nevertheless it is composed by the regional monsoons. I think it is important to stress on the manuscript the need to have both, as this would help merging the contribution from the different communities dedicated to the regional monsoons (actually this is done in some parts, I would check it to be consistent in the whole manuscript)*

**Response:**

Here is a clarification for the terms "global monsoon" and "global monsoons". . In the revised manuscript, we use "global monsoon" to highlight the consistent changes of all regional monsoons at longer time scales, and the role of the monsoon system in the global hydrological cycle; whereas we use "global monsoons" to highlight the regional features of different monsoons and the contribution of regional monsoon systems to the global hydrological cycle **(P2, L14-17)**.

*2 - In the Introduction the issue of the recent observed decrease in precipitation over India and the tendency of the coupled models to have increased precipitation when the atmospheric CO2 increases should be discussed (i.e. issues of thermodynamical versus dynamical changes in precipitation as discussed in Cherchi et al., 2011 and in Endo and Kitoh, 2014 for the different monsoon regions) - see references: Cherchi et al. (2011) Clim Dyn 37 83-101 doi:10.1007/s00382-010-0801-7; Endo and Kitoh (2014) GRL 41 1704-1710 doi:10.1002/2013GL059158.*

**Response:**

Thanks. This has been revised as suggested and the relevant papers have been cited **(P3, L14-18, L25-27)**.

*3 - Table 1: a useful information that should be added in this table and that should be mentioned in paragraphs 4.2 4.3 4.4 and 4.5 is the models involved in GMMIP that will be also involved in the other respective MIPs. This would help to know how many models (i.e. how large will be the sample) could be included in the comparison*

**Response:**

This is a good idea, but other MIPs do not provide the model information in their papers or websites, except for HighResMIP. We hope we can provide this information on the GMMIP website, pending the publication of CMIP6 documentation papers.

*4 - You should specify if you have specific requirements for the variables (and respective time-frequency) that should be saved as output from the GMMIP experiments (they should be listed in the manuscript)*

**Response:**

The variables and time frequency of model output are now listed in Appendix II **(P13-16)**.

*5 - You should specify what specific criterion should be used for the TIP-NSH experiment (tier-3) to cut off the sensible heating from the selected regions*

**Response:**

This has been clarified **(P7, L6-8; P25)**.

"…the vertical temperature diffusion term in the atmospheric thermodynamic equation at the bottom boundary layer is set to zero (Wu et al., 2012). The atmospheric component will not see the surface upward sensible heat flux (zero), whereas the land component is as usual."

*Some technical corrections:*

*Page 1, line 20: change "during" with "in"*

**Response:**

Corrected.

*Page 1, line 23: remove the comma after the word "DECK"*

**Response:**

In the CMIP6 framework, the "historical" simulation is not on the list of DECK experiments (see the overview paper of CMIP6, Eyring et al., 2016), so we have separated them with a comma.

*Page 1, line 27: I would use "benefit monsoons prediction .." instead of "benefit monsoon prediction .."*

**Response:**

Corrected.

*Page 2, lines 15-18: I think that in the Introduction the issue of internal feedback should be separated from that of external driven processes, and discussed in more detail*

**Response:**

We have re-organized the paragraphs **(P2-3).**

*Page 6, sections 3.2 and 3.3: what are the initial conditions for these experiments? how long are these experiments? I would suggest including these information also in the text not only in the table*

**Response:**

Added **(P6, L10-12)**.

*Page 6, line 15: I would insert "coupled" between "historical" and "climate"*

**Response:**

Corrected **(P6, L21)**.

*Page 6, lines 23-24: I think it is better to consider the tier-3 as a perturbation of the Tier-1 rather to the DECK (it is the same, it is just a matter of flow of the description)*

**Response:**

The Tier-3 experiment is to test the sensitivity of high topography and the sensible heating associated with it, mainly focusing on the effects on monsoon climatology. It is too expensive (and likely unnecessary) to run for the same period as GMMIP Tier-1 experiment.

*Page 7, line 5: what do you mean by "standard CMIP6 horizontal and vertical resolutions"*

**Response:**

This has been revised to "the same resolution as used in DECK".

*Page 7, section 4.1: the chain of comparisons between different experiments is a bit confusing. Consider rewriting the paragraph. More for the comparison of Tier-2 experiments with pre-industrial and historical simulations, please consider that in the former (tier-2 experiments) you have prescribed SST in selected regions but you have also the contribution of anthropogenic GHG and aerosols*

**Response:**

Related parts have been rewritten. The Tier-2 experiments are assumed to have "real" forcing signal and decadal drivers in the ocean. Thus comparing it with pre-industrial simulations would allow us to check the role of external forcings, while comparing it with historical run would allow us to check the roles of internal decadal modes, e.g., IPO and AMO **(P7, L22-28)**.

*Page 7, lines 28-29: not clear, please rewrite. Why high resolution in the mid-latitudes?*

**Response:**

This has been rewritten **(P8, L12)**.

*Page 7-8, section 4.3: It should be mentioned in the manuscript that in HighResMIP the SST used to build the AMIP experiments will be used as daily mean, differently from the other AMIP protocol. This should be considered also for the kind of comparison that would result. Also in HighResMIP the aerosols would be sort of prescribed (mandatory use of MPI simple plume module for anthropogenic aerosols). This should also be mentioned and discussed in terms of possible comparisons with GMMIP experiments*

**Response:**

These points have been clarified in the revision **(L8, P13-16)**.

*Page 8, lines 24-26: this could also be a hard comparison because of the specificity of the HighResMIP experiments as mentioned in the comment just above. You should mention what kind of specific metrics/analysis could be used/you have in mind for this comparison?*

**Response:**

Yes. The statement has been revised to "A comparison of CORDEX2 evaluation framework experiments forced with daily mean SST to HighResMIP Tier 1 runs over global monsoon domains will provide information on the similarities and differences of the added values derived respectively from high resolution global models and regional climate models." **(P9, L12)**

*Page 9, line 7: "ACGM" should be "AGCM"*

**Response:**

Corrected **(P9, L23)**.

*Page 9, lines 7-9: both 20CR and ERA20C are global atmospheric reanalyses that assimilate only the surface pressure (and the SST are prescribed)*

**Response:**

This has been revised **(P9, L23-24)**.

*Page 9, line 7-8: you should include references for 20CR and ERA20C*

**Response:**

References added **(P9, L24)**.

*Page 9, line 15: "global monsoon" instead of "global monsoons"*

**Response:**

As in the response to your first comment, here we use the term "global monsoons" to emphasize different monsoon domains.

*Page 11: "Data availability" should be an appendix, I guess (see also general comment above for specific requirements on variables and related time frequency)*

**Response:**

The location of "Data availability" is suggested by the CMIP6 special issue organizer. We added a part to show data requirements in the appendix II **(P13-16)**.

*Page 11, line 25: insert "coupled" between "historical" and "simulation"*

Response:

Done **(P12, L13)**.

**Reviewer 5: K. E. Taylor**

*The CMIP Panel is undertaking a review of the CMIP6 GMD special issue papers to ensure a level of consistency among the invited contributions, and to consider whether the co-chairs adequately addressed the key questions outlined in our request to submit a paper. We very much welcome the important contribution from GMMIP to the CMIP6 special issue, below are a few comments:*

*  - Thank you for responding to our request (through the editor) promising to revise your title consistent with others in this special issue.*

*  - It would be helpful to replace "other MIPs" (and similar constructs) in some places with "endorsed MIPs" to make clear the cohesiveness of the CMIP6 set.*

**Response:**

Revised as suggested.

*  - There is essentially no discussion of any special data required for your analyses.*

*  - Perhaps there are no special needs, but it would be good to at least include some discussion of the priority 1 data request. In particular, if data with temporal resolution greater than a month are needed, then indicate this, especially if 3-d (spatially) fields are required. Without emphasizing this, some groups may not realize the importance of saving the needed fields.*

**Response:**

We now have added Appendix II to clarify what kinds of data are required **(P13-16)**.

[revised manuscript text omitted]

---

## Author Response (AR2)

**In the following, the texts with italic font are the reviewer's original comments, and the texts with normal font are authors' response. The revised part in the manuscript is marked by red.**

Publish subject to minor revisions (Editor review) (25 Aug 2016) by Dr. Richard Neale

*Comments to the Author:*

*Concerns were raised about the design of the orographic and removed sensible heat flux experiments and they should put a little more clarification on this in the text.*

**Response:**

We have add some explanation on why we design the *orographic and removed sensible heat flux experiments* in the text **(P6, L27-30; P7, L9-11)**. And we add an appendix (Appendix II) to describe the details of the domain of modified orography and how to close the sensible heating **(P13, L16-26)**.

"The aim of the "orographic perturbation" is to understand quantitatively the regional response to the orographic perturbation from both the thermal and dynamical aspects. The results will be very helpful to understand the topography effect on the atmosphere and associated physical processes locally and quantitatively, such as the distribution, intensity, and frequency changes in the precipitation over wide monsoon regions."

"The sensible heat over the elevated topography is regarded as the main driver of the behaviour of the low level atmosphere and possibly also the upper troposphere and lower stratosphere (Wu et al., 2016). To examine the importance of elevated heating in monsoon from perspective of multi-model comparison …"